# Functional integration of a serotonergic neuron in the *Drosophila* antennal lobe

Xiaonan Zhang, Quentin Gaudry*

Department of Biology, University of Maryland, College Park, United States

**Abstract** Serotonin plays a critical role in regulating many behaviors that rely on olfaction and recently there has been great effort in determining how this molecule functions in vivo. However, it remains unknown how serotonergic neurons that innervate the first olfactory relay respond to odor stimulation and how they integrate synaptically into local circuits. We examined the sole pair of serotonergic neurons that innervates the *Drosophila* antennal lobe (the first olfactory relay) to characterize their physiology, connectivity, and contribution to pheromone processing. We report that nearly all odors inhibit these cells, likely through connections made reciprocally within the antennal lobe. Pharmacological and immunohistochemical analyses reveal that these neurons likely release acetylcholine in addition to serotonin and that exogenous and endogenous serotonin have opposing effects on olfactory responses. Finally, we show that activation of the entire serotonergic network, as opposed to only activation of those fibers innervating the antennal lobe, may be required for persistent serotonergic modulation of pheromone responses in the antennal lobe.

## Introduction

Serotonin (5-HT) is a ubiquitous neuromodulator that is found throughout phylogeny where it alters sensory (*Cornide-Petronio et al., 2015*; *Fields, 2004*; *Gaudry and Kristan, 2009*; *Yokogawa et al., 2012*), motor (*Lillvis and Katz, 2013*; *Schwartz et al., 2005*), and cognitive function (*Meneses and Liy-Salmeron, 2012*; *Sitaraman et al., 2008*; *Yuan et al., 2005*). Behaviors that rely critically on olfaction also depend on proper serotonin signaling (*Albin et al., 2015*; *Lent et al., 1989*; *Kravitz, 2000*; *Dierick and Greenspan, 2007*; *Becnel et al., 2011*; *Johnson et al., 2011*; *Ganesh et al., 2010*). Recently, there has been great interest in how this transmitter influences olfactory processing across model organisms including vertebrates (*Liu et al., 2012*; *Petzold et al., 2009*; *Ranade and Mainen, 2009*; *Kapoor et al., 2016*) and invertebrates (*Kloppenburg and Mercer, 2008*; *Gatellier et al., 2004*; *Dacks et al., 2008*, *2009*). To understand how serotonin affects olfactory processing, we must understand both how the modulator is released in relation to olfactory signals, and determine the cellular effect of the modulator on each neuron in the circuit. While progress has been made towards these goals, it still remains unclear how endogenous serotonin is released into olfactory circuits and how it shapes odor responses.

Serotonergic neurons in both mammals and insects show stimulus evoked responses (*Ranade and Mainen, 2009*; *Hill et al., 2002*; *Cohen et al., 2015*), but in no phylogenetic group have the odor responses been comprehensively mapped for serotonergic neurons that project to the first olfactory relay. Knowing how serotonin release correlates with olfactory sampling is critical for forming physiological models of 5-HT function in olfaction. For example, in mammals serotonin indirectly inhibits olfactory receptor neurons (ORNs) in the olfactory bulb (OB) (*Petzold et al., 2009*). Serotonin may thus serve as a gain control mechanism in the bulb if it is released during olfactory sampling, or it may actually boost olfactory responses should olfactory stimuli inhibit serotonergic fibers within the bulb(*Dugué and Mainen, 2009*).

*For correspondence: qgaudry@umd.edu

**Competing interests:** The authors declare that no competing interests exist.

*Drosophila* is an ideal model system to investigate the interaction of serotoninergic neurons and olfactory circuits because of its well-characterized anatomy (*Figure 1A*), genetic accessibility, and analogous organization to mammalian olfactory circuits. More importantly, as with several other insect species, only one pair of serotonergic interneurons termed the contralaterally-projecting serotonin-immunoreactive deuterocerebral interneurons (CSDns), project to the first olfactory relay, the antennal lobes (AL) (*Kent et al., 1987*; *Sun et al., 1993*; *Dacks et al., 2006*) (*Figure 1B*). In flies, mechanisms exist to label and manipulate this neuron (*Singh et al., 2013*; *Roy et al., 2007*), and recent studies have shown the CSDns to be directly involved in pheromone-mediated behaviors such as courtship (*Singh et al., 2013*). Here, we sought to (1) describe for the first time the olfactory receptive fields of a serotonergic neuron that innervates a primary olfactory structure, (2) characterize the synaptic integration of this neuron within the antennal lobe, and (3) determine how the endogenous release of serotonin influences olfactory and pheromone processing in insects. Our results demonstrate that these neurons, the CSDns, are predominantly inhibited by olfactory stimulation, and that this inhibition arises from reciprocal synapses formed directly within the AL. We show that the CSDn likely also releases the fast-acting neurotransmitter, acetylcholine (ACh), and that these two molecules produce different effects with opposing polarities and time courses on their downstream targets. Finally, we report that despite the CSDns being the only serotonergic neurons to project to the AL, robust modulation of glomeruli that respond to the male pheromone, 11-cis-vaccenyl acetate (cVA), is only observed when the entire serotonergic network is stimulated in unison, rather than sole activation of the CSDns.

## Results

### Olfactory stimuli inhibit CSDn spiking

We first performed whole-cell recordings from the CSDn to determine if odor stimulation could drive serotonin release into the antennal lobe in a fast and transient manner. Several odorants, such as ammonia and ethyl acetate, indeed elicited rapid and diverse responses in these cells (*Figure 1C*). However, a broad panel of odorants spanning several chemical classes reveals that most olfactory stimuli actually suppress firing in the CSDn (*Figure 1D*). This odor panel was selected such that most known ORNs classes were activated by at least one odor in the panel (*Hallem and Carlson, 2006*; *Silbering et al., 2011*). Additionally, the panel also included several ethologically relevant odors (*Dweck, 2015*; *Stensmyr et al., 2012*; *Dweck et al., 2013*; *Kurtovic et al., 2007*). We sorted the odorants according to the strength of the hyperpolarization that they induce in the CSDn. This sorting shows that esters, which are byproducts of fermentation, are particularly effective at inhibiting the CSDn (*Figure 1E*).

The recruitment of inhibition within and between glomeruli in the antennal lobe does not largely depend on odor identity, but rather depends on total ORN activation (*Hong and Wilson, 2015*; *Olsen et al., 2010*). If the CSDn is sensitive to this same source of inhibition, then the strongest odors in our panel should elicit the strongest inhibitory responses in these cells. Here, we define the strength of an odor to be the total olfactory receptor neuron activity as measured by a local field potential (LFP) from the antennae (*Schneider, 1957*) (*Figure 1F*). We measured such field potentials for a subset of our odor panel and found a strong correlation between antennal LFP amplitudes and the strength of inhibition onto the CSDn (*Figure 1G*). These results support the notion that inhibitory responses in the CSDn may arise from the well-described inhibitory circuits of the antennal lobe.

### Odor-mediated inhibition in the CSDn likely arises from local circuitry within the antennal lobe

Two inhibitory neurotransmitters released by local interneurons (LNs) have been identified in the AL; GABA and glutamate (*Liu and Wilson, 2013*; *Wilson and Laurent, 2005*; *Chou et al., 2010*; *Root et al., 2008*). Odor evoked inhibition of the CSDn is likely meditated in part by both transmitter systems, as bath application of their respective antagonists block this inhibition (*Figure 2A–D*). While GABA and glutamate antagonists block the odor-evoked inhibition of the CSDn, it is not certain if this inhibition arises at the level of the AL. Alternatively, downstream circuits, such as the mushroom bodies or lateral horn, may project to the CSDn to provide this inhibition. Thus, we tested for pre- and postsynaptic specializations in the CSDns' neurites within the AL. We expressed a

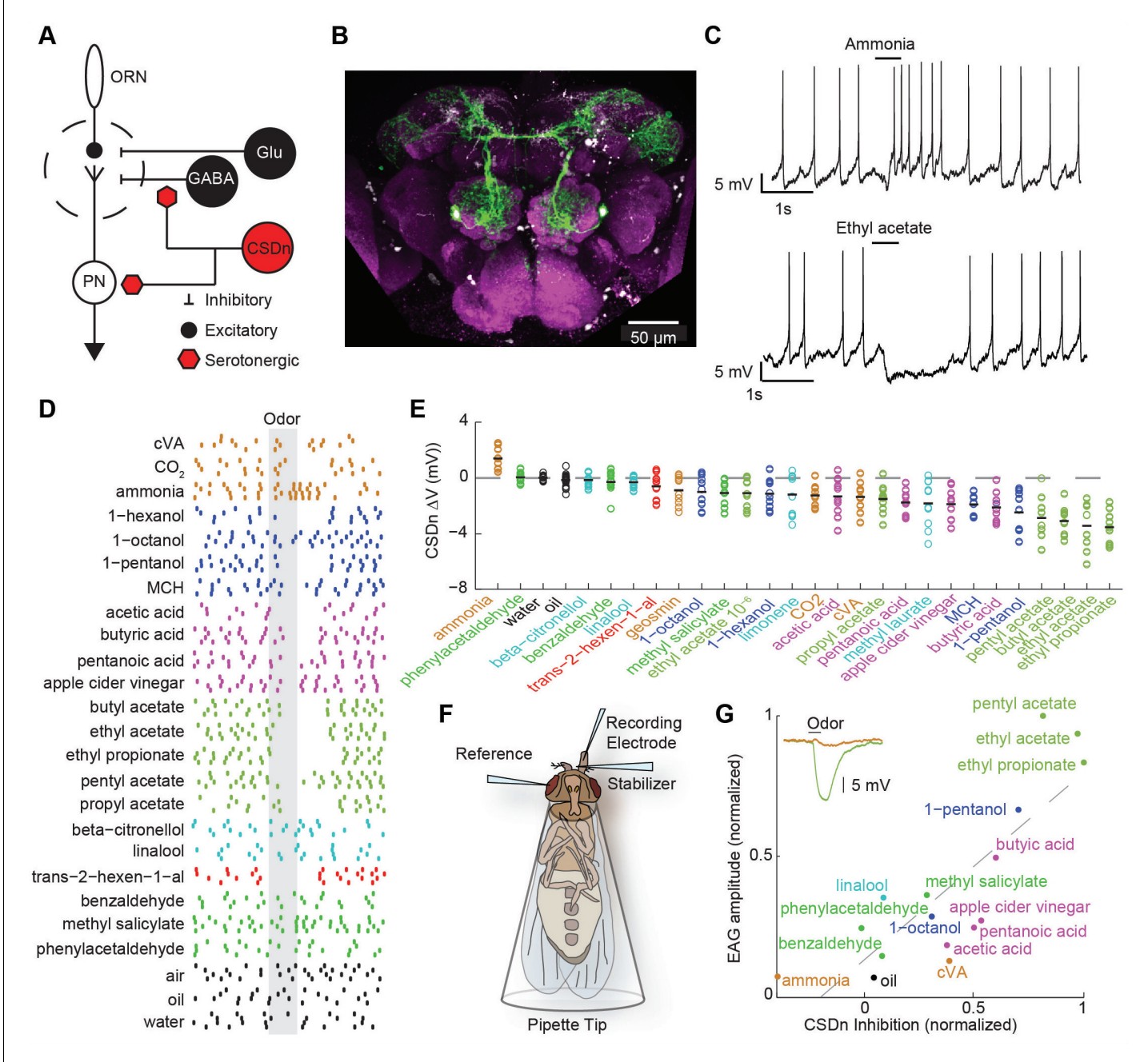

**Figure 1.** Olfactory stimulation hyperpolarizes serotonergic neurons innervating the AL. (**A**) Schematic representation of the AL circuitry showing excitatory connections from ORNs to PNs, and lateral inhibition from both GABAergic and glutamatergic interneurons. Serotonergic input onto LNs and PNs is inferred from previous studies across other model systems (see text). (**B**) An anterior to posterior Z-projection of a Drosophila brain expressing GFP in the R60F02-Gal4 (CSD -Gal4) promoter line to illustrate the innervation of the CSDn (green) in the antennal lobe (white-dashed circles). Serotonergic neurons are labeled with a 5-HT antibody and co-localize with the soma of the CSDn (white arrows). Neuropil (magenta) is labeled with the nc82 antibody. (**C**) Whole-cell recordings from a CSDn showing excitatory and inhibitory responses to odors. Horizontal black line denotes period of odor presentation (500 ms). (**D**) A raster plot from one experiment showing that most odors inhibit the CSDns. Each tick represents one action potential from a CSDn. Odors are grouped and colored according to chemical class. Ammonia, CO2, and cVA, which activate very few ORNs types are grouped together. All odors are diluted 100-fold in paraffin oil except cVA and methyl laurate, which are undiluted. (**E**) CSDn responses are sorted by increasing strength of hyperpolarization. Each open circle represents one preparation. Horizontal black bar is the mean of 10 preparations. (**F**) Schematic representation of EAG recording paradigm. (**G**). Regression analysis shows correlation between EAG responses and hyperpolarization of the CSDn. Insert shows sample EAG responses to ammonia and pentyl acetate. $R^2 = 0.69$, $p=0.00007$.

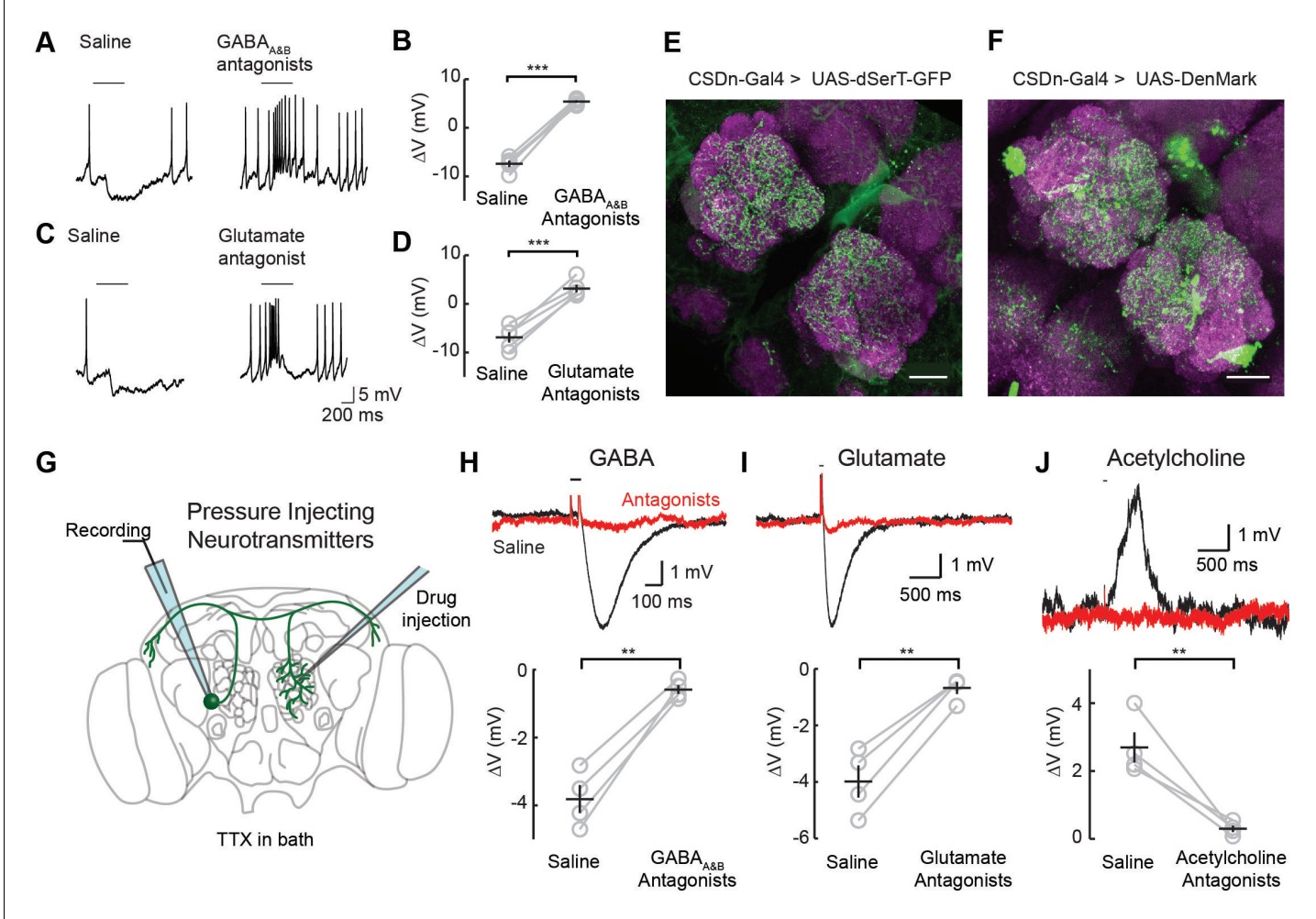

**Figure 2.** Inhibitory responses in the CSDn arise at the level of the AL. (A) Odor responses in the CSDn in normal saline and in the presence of the GABAA and GABAB receptor antagonists, picrotoxin (5 µM) and CGP54626 (50 µM) respectively. (B) GABA antagonists blocked the inhibition of the CSDn. n = 5, paired t-test, p=$5.59 \times 10^{-5}$. (C) The same protocol as in A except in the presence of 100 µM picrotoxin, which blocks inhibitory glutamatergic transmission in the fly. (D) Blocking inhibitory glutamate receptors also blocks the inhibtion of the CSDn. n = 5, paired t-test, p=$5.12 \times 10^{-4}$. (E) dSerT-eGFP is expressed selectively in the CSDn using the GAL4/UAS system. Presynaptic 5-HT release sites are seen as GFP signal in the AL. Neuropil is labeled as in *Figure 1A*. Scale bar = 20 µm in C and D. (F) The postsynaptic dendrite marker, DenMark, is expressed in the CSDn and visualized in green. (G) A cartoon representation of our protocol to reveal functional CSDn synapses within the AL. A sole CSDn neuron is drawn in green onto an schematic of the Drosophila brain. Cells are targeted using GFP and their neurites are stimulated with neurotransmitters delivered via pressure injection into the contralateral antennal lobe. To block polysynaptic and network contributions, TTX (1 µM) is added to the recording saline. (H) Top, sample hyperpolarizations of the CSDn in response to pressure injection of GABA in saline (black trace) and saline containing GABA antagonists (red trace). Antagonists as in A. The horizontal bar above the trace denotes the duration of pressure injection, and the coincident brief upward transient is an artifact from the opening and closing of the pressure injector's valve. (H) bottom. A summary of recordings with GABA injection. Each gray circle represents one preparation. Black horizontal line is the mean across preparations and black vertical line shows the SEM. n = 4, paired t-test, p=0.002. (I) Same as in H, but for the application of glutamate and 100 µM picrotoxin. n = 4, paired t-test, p=0.002. (J) Same as in H, but for the application of acetylcholine and mecamylamine (100 µM). n = 4, paired t-test, p=0.008. *p<0.05, **p<0.01, ***p<0.001, N.S. = not significant. Same symbols used in all figures.

The following figure supplement is available for figure 2:

**Figure supplement 1.** The CSDn expresses cellular markers of pre- and postsynaptic release sites throughout its neurites.

serotonin reuptake transporter fused to eGFP (dSerT-eGFP) in the CSDns (*Park et al., 2006*). This molecule localized in the AL confirming that the CSDns release 5-HT within this structure. To determine if the CSDns also have postsynaptic sites in the AL, we expressed the dendritic marker, Den-Mark (*Nicolai et al., 2010*). The positive labeling observed for both molecules suggests that the CSDn forms reciprocal connections within the antennal lobe (*Figure 2E and F*). Specifically, we found labeling for both markers throughout the AL suggesting that all glomeruli interact with the CSDn in a bidirectional manner to some extent. Notably, the anterior lateral glomeruli, including DA1 and VA1d, which both process pheromones, had the least staining for both markers.

We next tested if these putative postsynaptic sites of the CSDn in the AL are indeed functional. We performed whole-cell recordings from the CSDn while pressure injecting neurotransmitters onto the dendrites of the neuron in the contralateral AL (*Figure 2G*). Such recordings were performed in the presence of tetrodotoxin (TTX) to block action potentials (APs) and to eliminate any polysynaptic contributions. The CSDn's dendrites within the AL show sensitivity to GABA, glutamate, and acetylcholine (*Figure 2H–J*). Combined these data suggest that the CSDn is inhibited by odors proportional to the total number of ORNs activated, and that this inhibition functionally arises from local connections within the AL.

## The CSDn inhibits AL neurons via serotonin and excites AL neurons in an acetylcholinergic manner

The significance of the olfactory-mediated inhibition of the CSDn will ultimately depend on its output connectivity and impact on neurons within the AL. Our immunohistochemistry and previous EM studies (*Sun et al., 1993*) imply that the CSDn has presynaptic release sites within the AL. Additionally, serotonin has been shown to modulate GABAergic interneurons in the ALs of moths (*Kloppenburg and Hildebrand, 1995*) and flies (*Dacks et al., 2009*) suggesting possible synaptic connections between these cell types. To test for connections between the CSDn and GABAergic local interneurons (LNs) in the AL, we used the R60F02-Gal promoter line to express the red-light activated channelrhodopsin, Chrimson (*Klapoetke, 2014*) in the CSDn. This line strongly labels the CSDns with only a few additional processes seen in the subesophageal ganglion (*Singh et al., 2013*). We recorded the responses of randomly sampled LNs in the dorsal lateral cluster while stimulating the CSDn with a brief pulse of light (*Figure 3A*, *Figure 3—figure supplement 1*). Stimulation of the CSDn results in a brief depolarization of the LNs followed by a delayed hyperpolarization. (*Figure 3B*). Surprisingly, the potent 5-HT receptor antagonist, methysergide, is not effective at blocking this excitatory synapse (*Figure 3B,C*). This result suggests that the CSDn may release another neurotransmitter in addition to serotonin.

In flies, acetylcholine is the primary excitatory neurotransmitter in the nervous system, including the AL (*Buchner, 1991*; *Kazama and Wilson, 2008*), and therefore is a strong candidate for a co-transmitter. We applied the nicotinic receptor antagonist mecamylamine and found that it blocks the CSDn-mediated depolarization in the LNs (*Figure 3C*). The delayed hyperpolarization was not blocked by mecamylamine, but rather it was blocked by methysergide suggesting it is serotonin-mediated (*Figure 3D*). Depolarizing the LNs prior to CSDn stimulation demonstrates that this inhibition is indeed strong enough to prevent LN firing (*Figure 3E–G*). We conducted a larger survey of LNs using voltage-clamp recordings and found nearly identical results (*Figure 3—figure supplement 2*). These data suggest that the CSDn may release two neurotransmitters that have opposing roles in the AL; acetylcholine, which is predominately excitatory and serotonin, which broadly inhibited these GABAergic LNs. Consistent with this notion, we found that the CSDn is labeled by the ChAT-Gal4 promoter line, and that it is immunopositive for both ChAT and VAchT (*Figure 3—figure supplement 3*).

The synapses between the CSDn and the LNs were small and thus required us to elicit a barrage of action potentials in the CSDn in order to observe postsynaptic effects. Such stimulation is also likely to recruit polysynaptic pathways. We thus devised a new strategy called TERPS (Tetrodotoxin Engineered Resistance for Probing Synapses) to exclusively test for monosynaptic connections between neurons. We recorded from LNs in the presence of TTX to block all action potentials in the brain. We then selectively rescued spiking in the CSDn by co-expressing Chrimson and the TTX-insensitive sodium channel, NaChBac (*Ren et al., 2001*; *Nitabach et al., 2006*) (*Figure 3H*). Brief pulses of red light resulted in broad plateau potentials in CSDns expressing NaChBac and Chrimson (*Figure 3—figure supplement 4*). Such potentials were sufficient to elicit a brief depolarization and

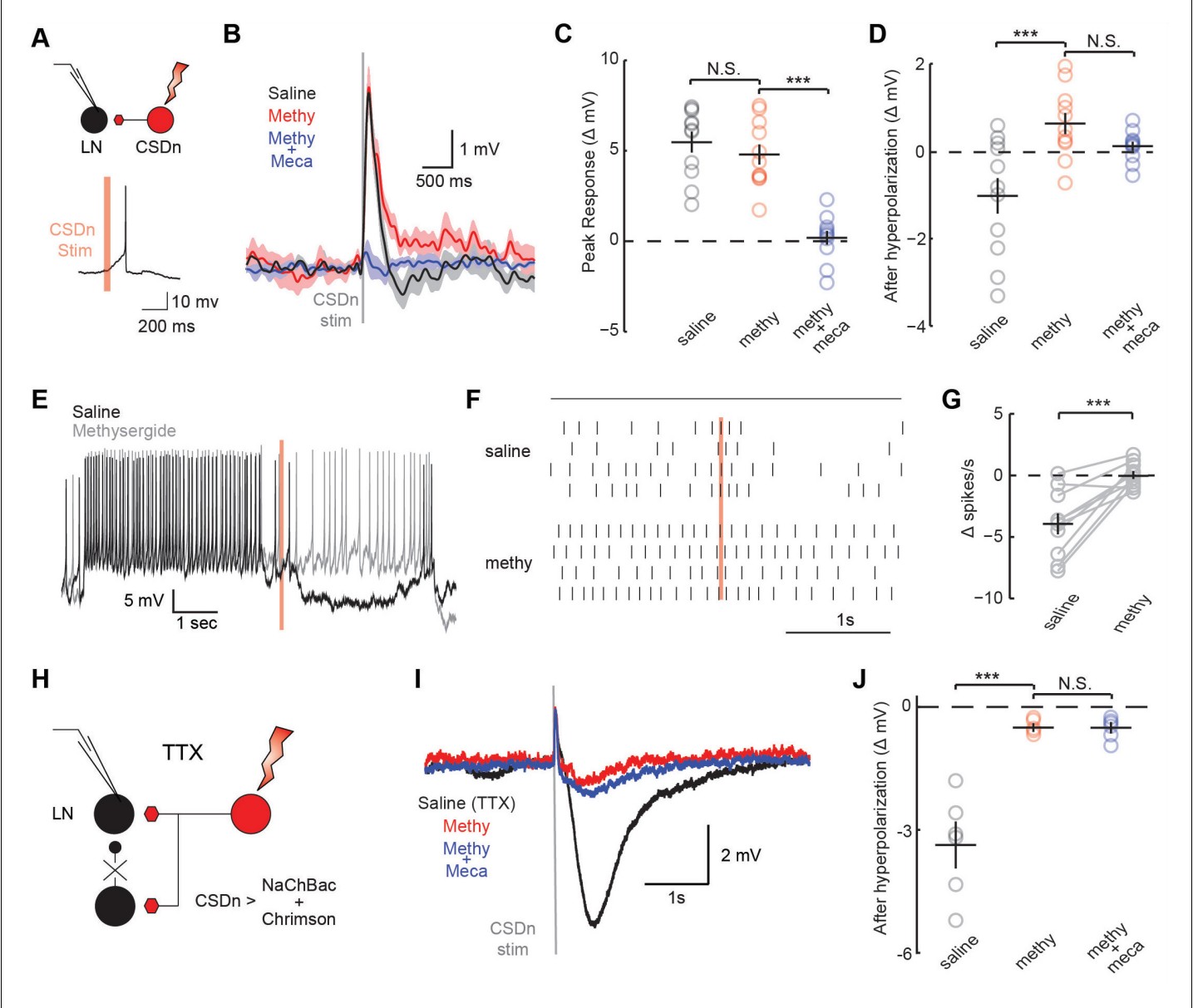

**Figure 3.** CSDn stimulation monosynaptically inhibits LNs and polysynaptically excites them. (**A**) Schematic representation showing optogenetic stimulation of the CSDn and whole-cell recording of GABAergic LNs. bottom, stimulation of CSDn results in an action potential in an LNs. (**B**) LNs were held at −60 mV and CSDn stimulation resulted in a fast depolarization followed by a delayed hyperpolarization. Methysergide (50 μM, red) bocked the delayed hyperpolarization but has no effect on the depolarization. Mecamylamine (100 μM, blue) blocked the depolarization. (**C,D**). Summary statistics for B. Methysergide has no effect on the peak depolarizing response, n = 11, ANOVA, F = 58.93, p=4.13 × $10^{-9}$, saline versus methysergide p=0.42. The addition of mecamylamine eliminated the depolarization from CSDn stimulation, methysergide versus methysergide plus mecamylamine p=9.4 × $10^{-8}$. Methysergide did block the delayed hyperpolarization, ANOVA, F = 11.01, p=0.0006. Saline vs methysergide p=5.02 × $10^{-4}$. Mecamylamine had no effect the hyperpolarization p=0.3403. (**E**) An LN was depolarized to −30 mV to magnify the CSDn evoked inhibition. This inhibition is blocked by methysergide. (**F**) A raster plot showing the inhibition of LN spikes. Black horizontal line above raster denotes period of depolarization to −30 mV. CSDn stimulation occurred during the 40 ms red bar. (**G**) Summary of such experiments at −30 mV. n = 10, paired t-test p=8.47 × $10^{-4}$. (**H**) Schematic representation of NaChBac experiments. NaChBac and Chrimson are co-expressed in the CSDn. TTX is used to block all action potentials in the brain except in the CSDn. (**I**) NaChBac potentials in the CSDn cause a fast depolarization and delayed hyperpolarization in the LNs. The hyperpolarization is blocked by methysergide (50 μM, red). Mecamylamine (200 μM, blue) did not block the fast depolarization. (**J**) Summary statistics for experiments in I. ANOVA, n = 6, F = 44.25, p=1.08 × $10^{-5}$, saline vs methysergide p=2.70 × $10^{-5}$, methysergide vs methysergide plus mecamylamine p=0.999.

The following figure supplements are available for figure 3:

**Figure supplement 1.** Demonstration and calibration of Chrimson activation of the CSDn.

*Figure 3 continued on next page*

*Figure 3 continued*

**Figure supplement 2.** Stimulation of the CSDn depolarizes LNs via acetylcholinergic transmission and subsequently inhibits LNs via serotonin.

**Figure supplement 3.** The CSDn may release acetylcholine as a co-transmitter.

**Figure supplement 4.** Co-expression of the TTX-insensitive sodium channel, NaChBac, and Chrimson can be used to effectively test mono-synaptic versus poly-synaptic connections.

a strong hyperpolarization in LNs (*Figure 3I*). The hyperpolarization was blocked by methysergide, but neither methysergide nor mecamylamine blocked the depolarization (*Figure 3J*). This depolarization is thus likely mediated by electrical coupling. Together these data show that the serotonergic connections from the CSDn to the LNs are likely monosynaptic, while the acetylcholinergic connections appear polysynaptic.

We performed a similar analysis on the connections between the CSDn and projection neurons (PNs, *Figure 1A*) sampled randomly (*Figure 4*, *Figure 4—figure supplement 1*). We found that the connections appear highly similar to those observed between the CSDn and LNs, only much smaller in amplitude. CSDn stimulation resulted in a brief depolarization of the PNs followed by a delayed hyperpolarization. TERPS analysis revealed that most of the excitatory connections onto the PNs are also polysynaptic while the serotonergic connections are likely direct.

## Endogenous serotonin suppresses PN responses to odor

The inhibitory effect of serotonin released from the CSDn is surprising given that previous studies in insects have shown that exogenous 5-HT boosts olfactory responses in both LNs and PNs (*Dacks et al., 2008*, *2009*; *Kloppenburg et al., 1999*). In *Drosophila*, such studies strongly emphasized modulation of the DA1 glomerulus (*Dacks et al., 2009*; *Singh et al., 2013*) because of its sensitivity to the male pheromone cVA, which is critical for normal courtship behavior (*Kurtovic et al., 2007*; *Dickson, 2008*; *Kohl et al., 2015*). We thus sought to test the effects of manipulating endogenous serotonin release on olfactory responses in the DA1 glomerulus. We first employed a pharmacological approach because it allows us to easily manipulate serotonergic transmission without altering Ach release from the CSDn. To reduce serotonergic transmission, we blocked postsynaptic receptors by adding the broad serotonin antagonist, methysergide to our recording saline (*Figure 5A*). Blocking serotonin receptors increased the odor responses of PNs innervating this glomerulus (*Figure 5B and E*). This suggests that serotonin functions naturally to suppress DA1 responses and that methysergide removes this suppression. To elevate serotonin levels, we used the selective serotonin re-uptake inhibitor, fluoxetine (*Figure 5C*). Unlike the exogenous application of 5-HT, fluoxetine should specifically concentrate serotonin at naturally occurring release sites (*Calviño and Szczupak, 2008*; *Calviño et al., 2005*). Consistent with an inhibitory function for serotonin, fluoxetine decreased DA1 PN odor responses (*Figure 6D*).

We tested two additional glomeruli DM6, and DL5, that respond to a broad range of non-pheromone odorants (*Hallem and Carlson, 2006*). In all three glomeruli that tested, we found that odor responses diminished as serotonergic transmission increased (*Figure 5E–G*). These results are surprising considering that a previous study, which used calcium imaging, found the exact opposite effect on the DA1 glomerulus with bath application of 5-HT (*Dacks et al., 2009*). To rule out differences in methodologies as an explanation for these observations, we tested the effect of exogenous 5-HT on odor responses. We show that exogenous serotonin reliably enhanced odor responses despite the finding that manipulating endogenous transmission suppresses odor responses (*Figure 5H–J*).

## DA1 odor responses are not modulated via the CSDn

The ultimate test of serotonin's role in shaping odor responses should come from measuring PN odor responses while directly manipulating endogenous release through the activity of serotonergic neurons themselves. We thus measured DA1 PN odor responses while stimulating the CSDns with

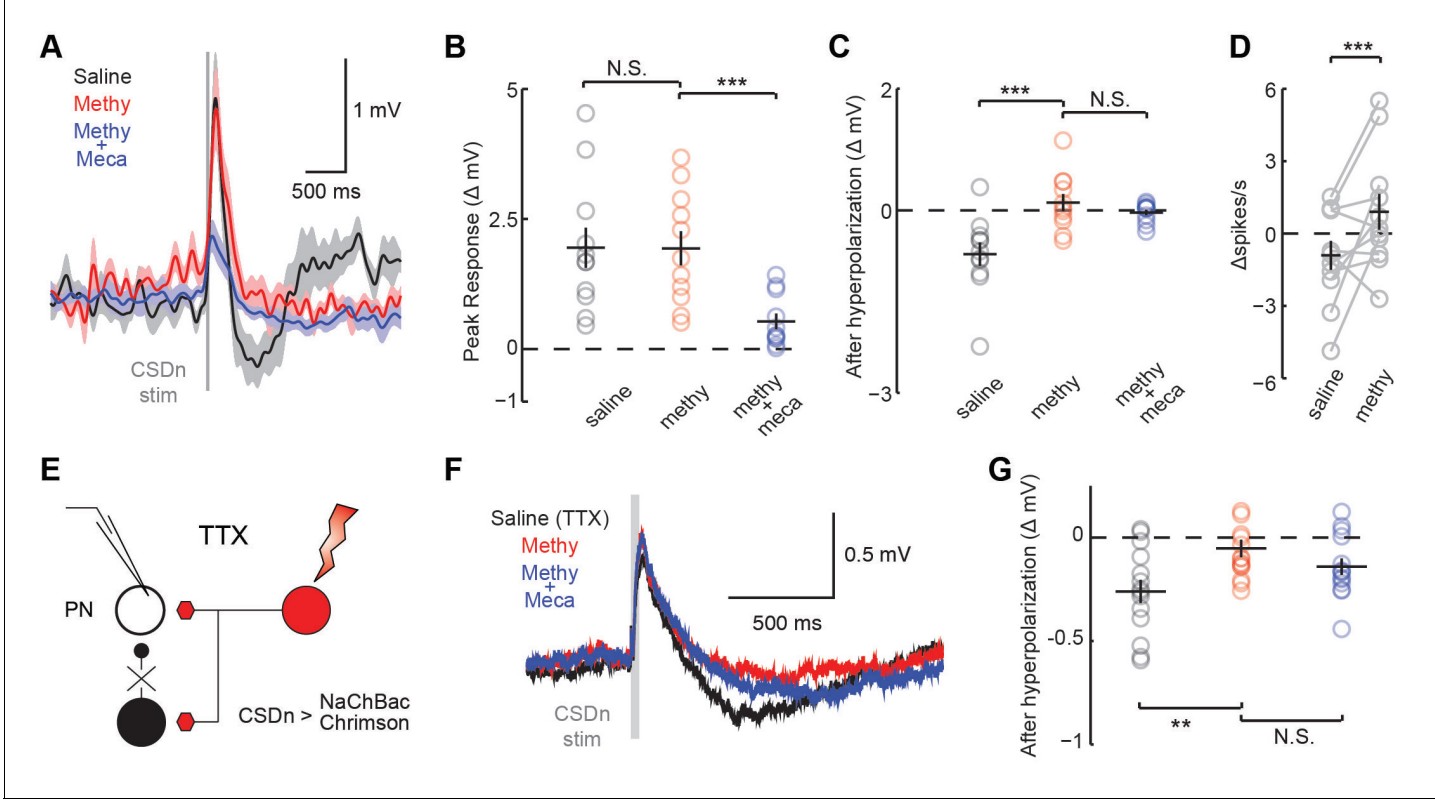

**Figure 4.** CSDn makes similar connections onto PNs as LNs. (**A**) PNs were held at −60 mV. Stimulation of the CSDn depolarizes PNs briefly and results in a delayed hyperpolarization (gray trace, saline). (**B**) The early depolarization could not be blocked by methysergide (50 μM) but was blocked by mecamylamine (100 μM), ANOVA, n = 11, F = 27.6, p=1.77 × 10$^{-6}$, saline vs methysergide p=0.99, methysergide vs methysergide plus mecamylamine p=8.67 × 10$^{-6}$. (**C**) The delayed hyperpolarization was fully blocked by methysergide, while mecamylamine had no further effect on the delayed part of the response, ANOVA, n = 11, F = 13.32, p=0.0002, saline vs methysergide p=2.70 × 10$^{-4}$, methysergide vs methysergide plus mecamylamine p=0.63. (**D**) PNs were depolarized to −30 mV to induce spiking and to amplify the effects of the hyperpolarization. At −30 mV CSDn stimulation significantly reduced PN firing, n = 11, p=0.031. (**E**) PNs were patched in saline containing TTX to block all activity in the brain. NaChBac and Chrimson were co-expressed in the CSDn to selectivity restore activity only in this neuron to probe monosynaptic connections with randomly selected PNs. (**F**) CSDn stimulation rapidly depolarized the PNs and then hyperpolarized them in TTX. (**G**) The hyperpolarization was blocked by methysergide, ANOVA, n = 14, F = 5.58, p=0.0096, saline (TTX) vs methysergide p=0.0072, methysergide vs methysergide plus mecamylamine p=0.35.

The following figure supplement is available for figure 4:

**Figure supplement 1.** Stimulation of the CSDn results in a fast acetylcholine-dependent inward current and a delayed serotonin-mediated outward current.

both optogenetic and natural olfactory stimuli. We first expressed Chrimson in the CSDns and excited them with a 10-Hz sine wave of red light. We briefly interrupted this stimulation to present cVA to the fly and measure DA1 responses (*Figure 6A*). This chronic activation of the CSDn did not significantly modulate DA1 responses to cVA (*Figure 6B and C*). We also attempted to induce CSDn-mediated modulation by driving activity in the CSDn with a natural olfactory stimulus, ammonia. We interleaved presentations of cVA with pulses of ammonia while recording from DA1 PNs (*Figure 6D–F*). This odor-evoked activity in the CSDn also failed to modulate DA1 odor responses. These results are surprising given that the DA1 glomerulus is sensitive to serotonin pharmacology (*Figure 5*).

## The DA1 glomerulus is sensitive to serotonin release for neurons outside the AL

How can serotonin alter DA1 odor responses if the only serotonergic neurons that project to the AL fail to modulate them? One possibility is that serotonergic modulation requires correlated activity

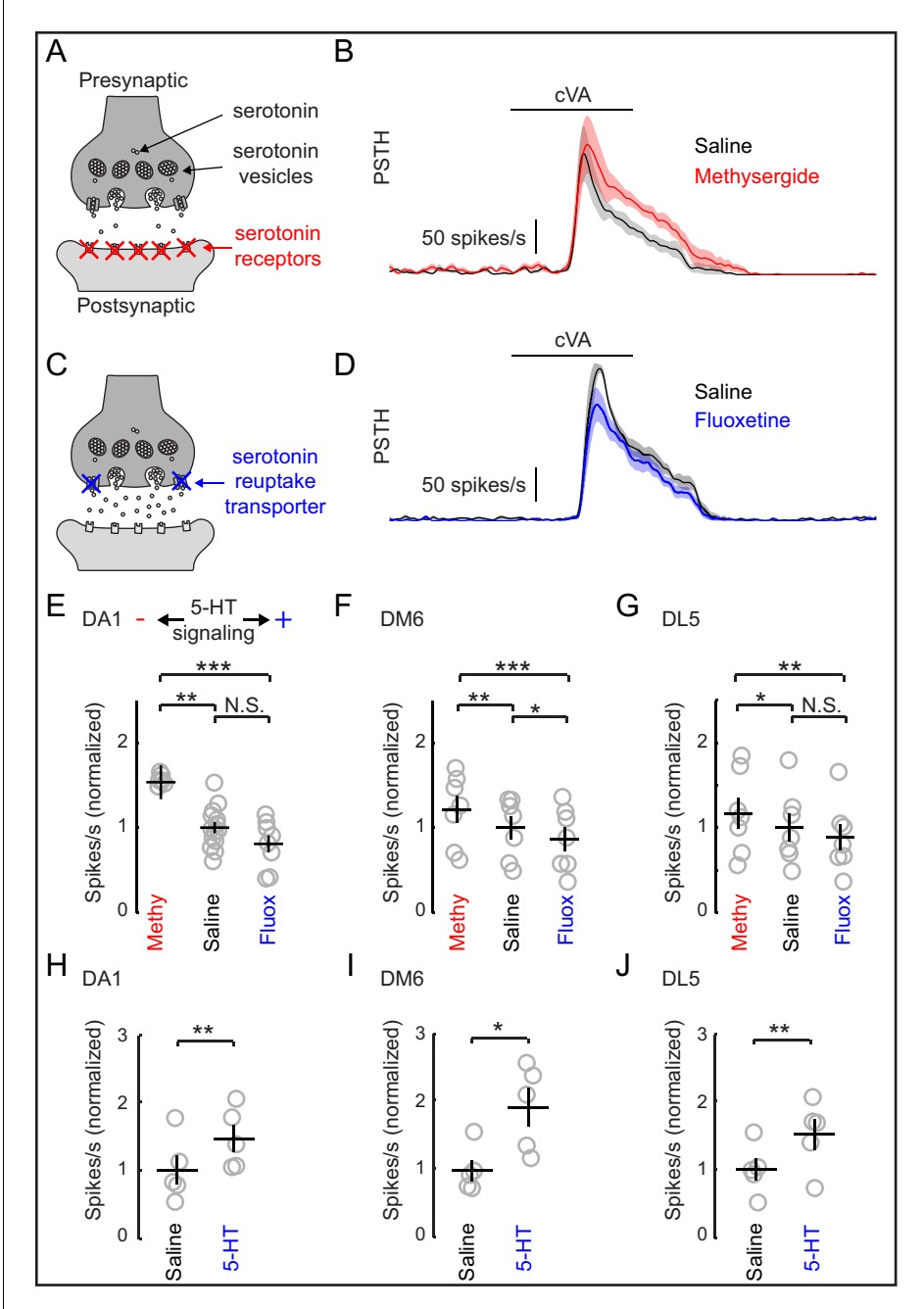

**Figure 5.** Increasing serotonergic transmission decreases PN responses in vivo. (**A**) A schematic of a serotonergic synapse showing vesicles and postsynaptic receptors. The receptors are blocked by the antagonist methysergide. (**B**) A mean PSTH of the DA1 PN responses to a 500 ms pulse of cVA in saline and methysergide. The shaded regions show the standard error of the mean. (**C**) A schematic representation of a serotonergic synapse showing serotonin reuptake transporters blocked by fluoxetine (10 μm). Blockade of reuptake transporters concentrates 5-HT in the synaptic cleft. (**D**) DA1 PN responses to a 500 ms pulse of cVA in saline and fluoxetine. (**E**) Quantification of DA1 responses. Data are normalized to the mean of the responses in saline. Serotonergic transmission increases from left to right (methysergide, saline, fluoxetine). n = 6 for saline vs. methy and n = 8 for saline vs fluox. ANOVA, p=$8.7 \times 10^{-4}$, F = 9.46. Tukey-Kramer post-hoc test was used for panels **E,F**, and **G**. Methy vs saline p=0.0055, saline vs fluox p=0.38, methy vs fluox p=0.0008. (**F**) DM6 PN responses to valeric acid ($10^{-6}$) under the same protocol. n = 7 for each condition, repeated measures ANOVA, p=$3.5 \times 10^{-5}$, F = 27.10. Methy vs saline p=0.0021, saline vs fluox p=0.036, methy vs fluox p=$3.0 \times 10^{-5}$. (**G**) DL5 PN responses to trans-2-hexenal ($10^{-7}$) under the same protocol. n = 7 for each condition, repeated measures ANOVA, p=0.0025, F = 10.32. Methy vs

*Figure 5 continued*

saline p=0.049, saline vs fluox p=0.20, methy vs fluox p=0.0019. (**H,I,J**) PN responses from the same three glomeruli are compared in saline vs exogenous serotonin application (104 M), n = 5 for each glomeruli. DA1 p=0.005, DM6 p=0.019, DL5 p=0.008.

The following figure supplement is available for figure 5:

**Figure supplement 1.** Exogenous application of 5-HT ($10^{-4}$) boosts PN responses to odors.

from the broader serotonin network. Most studies of serotonergic modulation of the mammalian OB stimulate the entire raphe and not just those axons projecting to the bulb itself (*Petzold et al., 2009*; *Kapoor et al., 2016*). We performed similar experiments in flies by using a *tryptophan hydroxylase* (Trh) Gal4 (*Alekseyenko et al., 2010*) promoter line to express Chrimson in all serotonergic neurons. We stimulated this population strongly for 10 min using the same 10 Hz sine wave

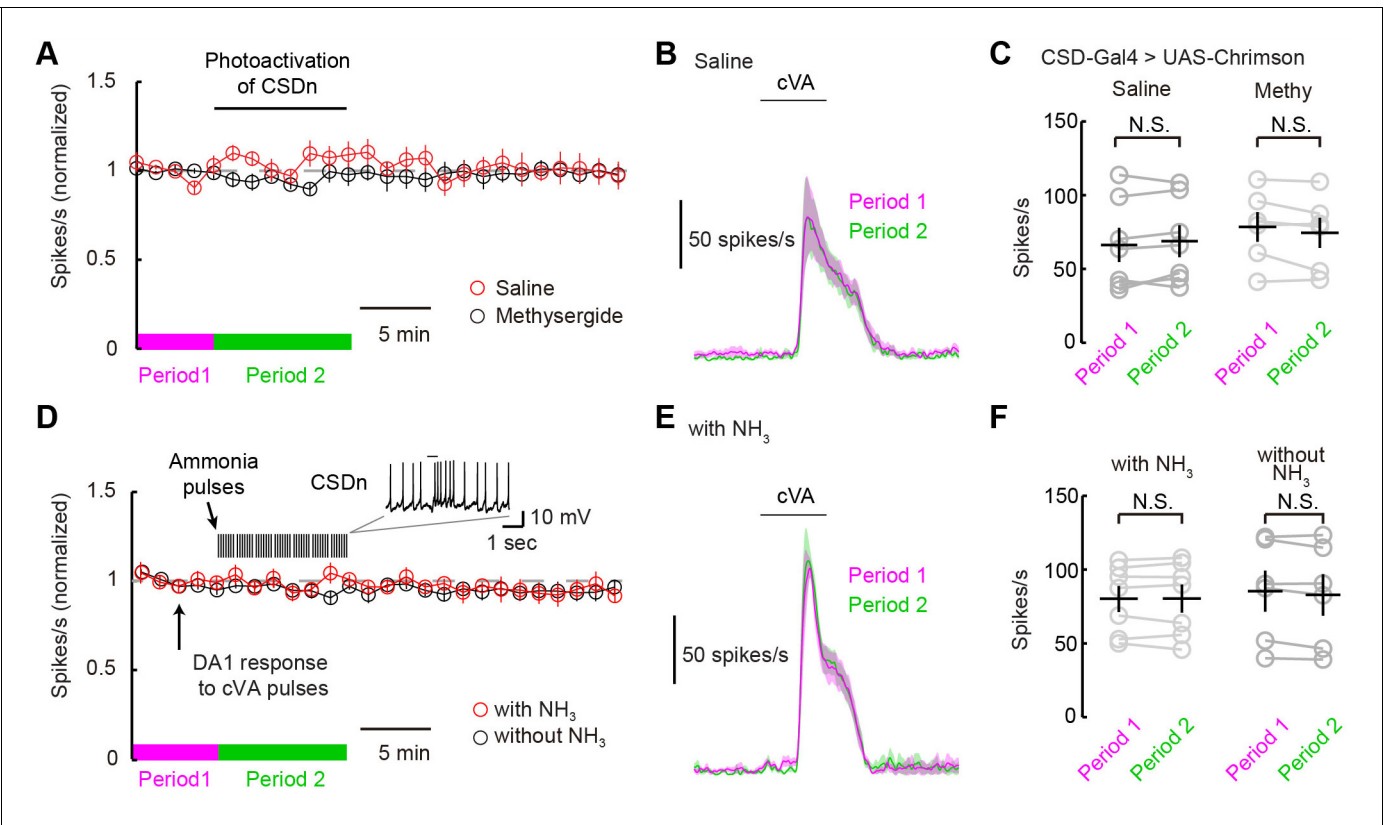

**Figure 6.** The CSDn does not modulate DA1 odor responses. (**A**) DA1 odor responses to cVA with chronic optogenetic stimulation of the CSDn (CSDn-Gal4 > UAS-Chrimson) in saline (red circles) and methysergide (black circles). The CSDn was continuously activated with a 10 Hz sine wave at 660 nm. The sinewave was intrupted temporarily to test DA1 responses to cVA. (**B**) PSTH showing DA1 spiking response to cVA before (magenta) and during (green) CSDn stimulation in saline. (**C**) Summary statistics for data in A and B. DA1 responses to cVA were not statistically modulated by optogentically driven CSDn activity. Period 1 vs Period 2 in saline. n = 7, p=0.307. DA1 odor responses also did not change with CSDn stimulation in methysergide. n = 6, p=0.138. (**D**) DA1 odor responses were sampled at 80 s intervals and the CSDn was stimulated with pulses of ammonia every 10 s (red circles). Ammonia was not presented simultaneoulsy with cVA to avoid fast lateral inhibition and excitation. DA1 resonses to cVA were stable in the absense of intermittent ammonia stimulation (black circles). Period 1 represents timeframe before ammonia stimulation in experimental group and Period 2 represents time frame during ammonia stimulation. (**E**) PSTH showing DA1 spiking response to cVA before (magenta) and during (green) ammonia presentation. (**F**) Summary statistics for data in D and E. DA1 responses to cVA were not statistically modulated by ammonia driven CSDn activity. n = 7, Period 1 vs Period 2 with NH3 presentation. p=0.976, paired t-test. DA1 odor responses also did not change without ammonia stimulation. n = 6, p=0.105, paired t-test.

of red light (*Figures 7A–C*). Activation of the whole serotonergic system produced a long-lasting suppression of odor responses. First, these results are consistent with the role of endogenous 5-HT in flies being to suppress odor responses rather than to boost them as seen with exogenous

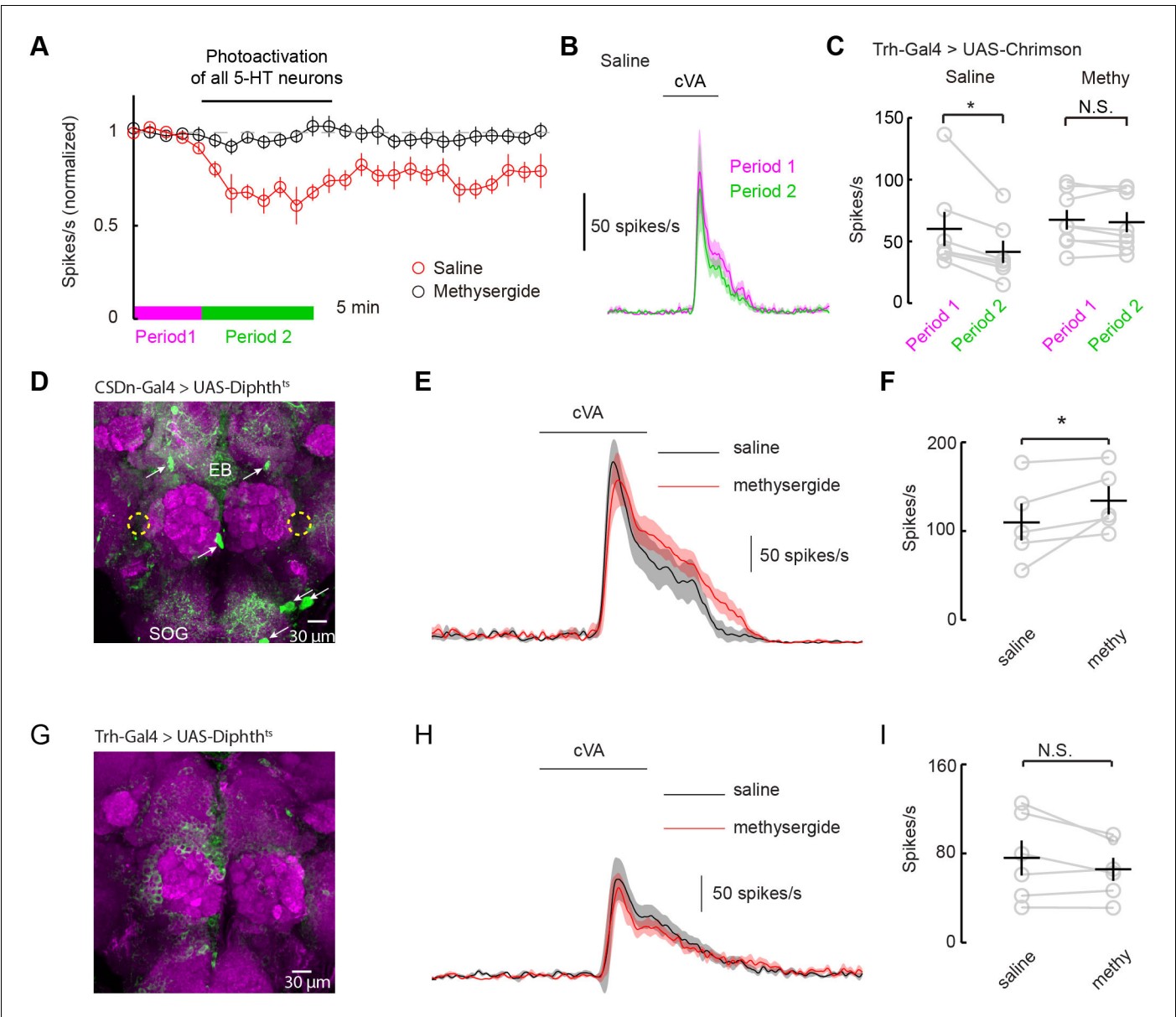

**Figure 7.** Serotonergic modulation of DA1 is governed by the network of serotonergic neurons and not the CSDn exclusively. (**A**) Optogenetic stimulation of all 5-HT neurons suppresses DA1 odor responses in saline (red) but not methysergide (black). (**B**) PSTH showing DA1 spiking response to cVA before (magenta) and during (green) Trh stimulation in saline. Odor pulse is 500 ms for **B,E**, and **H**. (**C**) Summary statistics of data in **A**. Stimulation in saline conditions reduced DA1 odor responses, n = 7 p=0.014, paired t-test. Methysergide blocked the suppression seen in normal saline n = 8, p=0.413. (**D**) The CSDn is killed by the expression of a temperature-sensitive variant of diphthera toxin. The expected location of the CSDn is illustrated with yellow, dashed circles. The remaining 5-HT circuit remains intact. 5-HT positive soma are indicated with white arrows. Note the 5-HT fiber innervation of the subesophageal ganglion (SOG) and the ellipsoid body (EB). (**E**) PSTH's of DA1 responses to the odor cVA in flies without CSDns. Responses in normal saline are shown in black and in the presence of methysergide in red. (**F**) Summary of DA1 responses from **E**. p=0.037, n = 5, paired t-test. (**G**) As in **D**, but with expression mediated by the Trh promoter to target dipthera toxin in all 5-HT neurons. The imaging gain was elevated until a signal in the 5-HT channel was discerned giving rise to a visible background level. Note the lack of clear dense green labeled neurons. Staining in the AL, the SOG, and the EB is largely absent. (**H**) Odor responses of flies with killed 5-HT systems. Colors and scales bars are the same as those in **E**. (**I**) A summary of DA1 responses from brains with ablated serotonergic systems, n = 6, paired t-test. p=0.172.

application. Second, they suggest that 5-HT from sources other than the CSDn may contribute to the modulation of DA1 odor responses.

Serotonergic neurons outside the AL may contribute to the modulation of DA1 neurons by either facilitating the CSDn's actions or by working independently of the CSDn. One mechanism by which serotonergic neurons other than the CSDn could influence AL circuits is by aiding in establishing a basal level of serotonin concentration in the fly haemolymph. In this model, all serotonin cells would contribute to tonic levels of circulating 5-HT and the DA1 would be sensitive to this serotonin. Alternatively, serotonin could modulate higher-order olfactory neurons that project back to the DA1 glomerulus. If either of these theories is true, then killing the CSDn should leave DA1 sensitive to serotonergic pharmacology. To test this theory, we expressed a temperature-dependent diphtheria toxin in the CSDns to kill them three days post eclosion (*Han et al., 2000*) (*Figure 7D*). This manipulation eliminated all 5-HT immunopositive processes in the antennal lobe, confirming that the CSDns are the only serotoninergic neurons to project to the AL in *Drosophila* (*Dacks et al., 2006*; *Singh et al., 2013*). Even in the absence of viable CSDns, the application of methysergide potentiated PN responses in the DA1 glomerulus (*Figure 7E and F*), suggesting that serotonergic neurons outside the AL contribute in modulating PN responses. We also expressed diphtheria toxin in all 5-HT neurons using the Trh promoter (*Figure 7G*). In this case, no potentiation was seen suggesting the boosting by methysergide was not caused by off target effects (*Figure 7H and I*). These findings are consistent with the notion that the CSDn is likely not the serotonergic neuron responsible for the pharmacological effects seen in the DA1 glomerulus.

## The CSDn modulates VA1d odor responses via Ach

We next wanted to test if the CSDns lack of ability to modulate pheromone responses in the AL applied similarly for glomeruli other than DA1. We thus repeated our experiments on the VA1d glomerulus, which responds to the female derived odor methyl laurate and whose cognate ORNs are necessary and sufficient for attraction in both males and females (*Dweck, 2015*). As with DA1, we saw no modulation of VA1d PN odor responses during chronic Chrimson activation of the CSDn (*Figure 8A and B*). In addition to chronic stimulation, we also applied brief pulses of light to drive the CSDn transiently during the peak of VA1d odor responses (*Figure 8C*). This protocol revealed a small but significant increase in the VA1d responses during CSDn stimulation. This effect was observed only during the portion of the odor response in which the CSDn was being activated (Figure D and E). In total, CSDn activation added about 1 action potential per PN per odor response in this glomerulus. Because CSDn stimulation boosted rather than suppressed VA1d responses, it is more likely to be caused by acetylcholine from the CSDn rather than serotonin. Indeed, this small boost in VA1d firing was observed in the presence of methysergide, further suggesting it is not mediated by 5-HT (*Figure 8F*). We repeated this brief stimulation protocol on the DA1 glomerulus, where we found no evidence of modulation (*Figure 8*, *Figure 8—figure supplement 1*). These results suggest that the CSDn may not contribute heavily to the modulation of pheromones in flies.

## Discussion

We capitalized on the fact that only one pair of serotonergic neurons project to the *Drosophila* AL to examine how such neurons integrate into primary olfactory structures and regulate pheromone processing. Ultimately, this analysis revealed unforeseen complexities regarding the function of serotonin these olfactory circuits. First, we found that the CSDns receive strong olfactory-mediated inhibition arising from connections within the AL. Second, we used immunohistochemistry to show that these neurons likely release acetylcholine as a co-transmitter and that acetylcholine and serotonin have opposing effects on postsynaptic cells in the AL. Third, we used pharmacological and optogenetic approaches to show that exogenous and endogenous 5-HT differentially effect odor responses. Fourth, we showed that while the CSDn is the only serotonergic centrifugal neuron to innervate the antennal lobe, it likely does not contribute to the modulation of pheromone processing odor channels. Strong serotonin-mediated modulation of the DA1 glomerulus is only observed by activation of the entire 5-HT network.

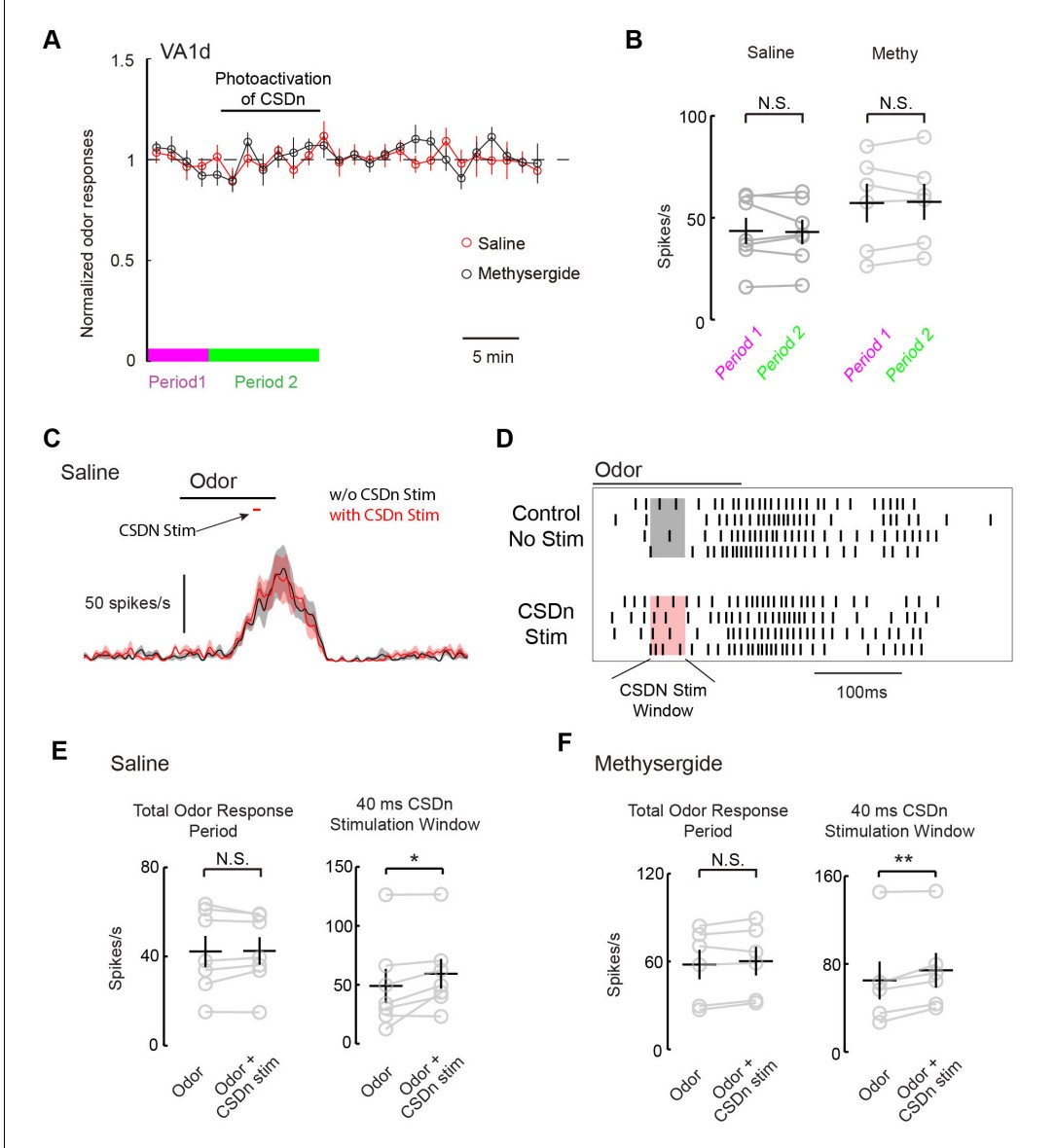

**Figure 8.** Acute stimulation of the CSDn alters VA1d responses via cholinergic transmission. (**A**) VA1d odor responses to methyl laurate were sampled every 80 s before, during, and after stimulation of the CSDn with Chrimson. Stimulation consisted of a 10 Hz sine wave that was interrupted briefly to sample VA1d response. (**B**) Chronic stimulation of the CSDn did not significantly alter VA1d responses either in saline (n = 7, p=0.77) or methysergide (n = 6, p=0.78). (**C**) A mean PSTH of VA1d odor responses with (red trace) and without (black trace) simulatneous CSDn activation with Chrimson (40 ms). (**D**) A raster showing an increase in the odor respnse of VA1d PNs during brief CSDn stimulation. Methyl laurate was presented undiluted. (**E**) In saline, PN firing over the entire duration of the odor response was unchanged (left, n = 7, p=0.84) but during the 40 ms of CSDn stimulation, VA1d responses increased (right,, n = 7, p=0.046). (**F**) In methysergide, PN firing over the entire duration of the odor response was also unchanged (left, n = 6, p=0.16). During the 40 ms of CSDn stimulation, VA1d responses increased (right, n = 6, p=0.008).

The following figure supplement is available for figure 8:

**Figure supplement 1.** Acute stimulation of the CSDn does not modulate DA1 odor respones.

## Serotonergic modulation of PN responses in *Drosophila*

We found that the predominant role of serotonin in the *Drosophila* AL is to suppress PN odor responses and this effect is seen across PNs that innervate different glomeruli. We observed suppression in glomeruli both sensitive to pheromones and food-derived odors, suggesting a common

theme across olfactory channels. Such global suppression could serve as a gain control mechanism to prevent PN responses from saturating or to allow signals from glomeruli less sensitive to serotonin to be boosted relative to other glomeruli (*Petzold et al., 2009*; *Dugué and Mainen, 2009*). Currently it is unclear if serotonin-insensitive glomeruli exist in the fly AL as all of the PNs that we sampled showed inhibition to serotonin.

Interestingly, previous studies in moths and *Drosophila* have suggested the opposite role for serotonin in modulating insect olfaction (*Dacks et al., 2009*; *Kloppenburg et al., 1999*). In these studies, the application of exogenous serotonin results in the boosting of olfactory responses in PNs. We attribute this difference to the mode of application, namely exogenous versus endogenous release. The influence of exogenous 5-HT is known to be concentration dependent, and low concentrations of 5-HT can suppress odor responses in PNs in moths (*Kloppenburg and Hildebrand, 1995*). However, it is unlikely that concentration alone explains our phenomenon since even strong optogenetic activation of all 5-HT neurons elicits suppression of PN odor responses in our experiments.

An alternative explanation might be that multiple classes of 5-HT receptors within the antennal lobe each have unique kinetics, affinities, and ability to excite or inhibit neurons. For example, 5-HT7 (*Becnel et al., 2011*) and 5-HT1B (*Yuan et al., 2005*; *Singh et al., 2013*) receptors are both found in the AL of the fly, but the 5-HT7 receptor is fully saturated before the 5-HT1B receptor begins to show any sensitivity to 5-HT (*Gasque et al., 2013*). Tonic bath application of serotonin might also preferentially activate one class of receptors, while phasic release from natural stores might favorably activate another class. Our data is consistent with these hypotheses. For example, we found that transient release of serotonin from the CSDn elicited only a brief excitatory inward current in LNs and PNs and a more prolonged outward inhibitory current (*Figure 3*, *Figure 3—figure supplement 2* and *Figure 4*, *Figure 4—figure supplement 1*). However, bath application of 5-HT resulted in a prolonged inward current that lasted the duration of 5-HT exposure (*Figure 5—figure supplement 1*). Conflicting and opposing effects of serotonin have also been reported in other systems. In crayfish, the rate of bath application of serotonin determines whether it will facilitate or depress synaptic input onto command neurons that trigger escape behaviors (*Teshiba et al., 2001*). Thus, the rate of change in serotonin concentration at CSDn to LN and PN synapses may be another reason why we observe differences in the modulation of odor responses with endogenous release and bath application.

What is the source of serotonin that mediates the suppression of olfactory responses in the pheromone-sensitive DA1 glomerulus? Interestingly, we found that DA1 is highly sensitive to serotonergic pharmacology, yet largely insensitive to serotonin from the CSDn. In fact, DA1 sensitivity to serotonergic pharmacology remains even when the CSDn is ablated. This lack of DA1 sensitivity to CSDn-derived serotonin is consistent with the finding that this glomerulus is only weakly innervated by the CSDn (*Singh et al., 2013*), but is still surprising given that the CSDn is the only serotonergic neuron that innervates the AL (*Kent et al., 1987*; *Sun et al., 1993*; *Dacks et al., 2006*). We propose that one of two mechanisms might allow stimulation of the entire 5-HT network to strongly modulate DA1 odor responses where CSDn stimulation cannot. First, serotonin released from Trh-Gal4 stimulation may modulate cells outside the AL that are inaccessible to the CSDn and that ultimately feedback into the AL. Such a model could extend to most of the pheromone sensitive glomeruli in the antennal lobe as the VA1d glomerulus is also weakly innervated by the CSDn and shows weak modulation due to CSDn stimulation. Second, stimulation of Trh-Gal4 may elevate serotonin in the AL through paracrine signaling to levels higher than can be achieved by the CSDn alone. This paracrine or hormonal function for serotonin may not be so surprising considering serotonin can be readily measured in the haemolymph of many invertebrates (*Panksepp et al., 2003*; *Tecott, 2007*; *Lange et al., 1989*), and neuronal activity alone is sufficient to raise circulating 5-HT in invertebrate models (*Levenson et al., 1999*). In the fly larvae, a single 30 s bout of activity in serotonergic neurons can drastically elevate 5-HT levels for several minutes as measured extracellularly by fast-scan cyclic voltammetry (*Borue et al., 2009*). At these concentration levels, 5-HT may be able to diffuse long distances within the fly nervous system.

## The anatomy, connectivity, and function of the CSDn

While our experiments have revealed a new role for serotonin in modulating olfactory responses in insects, the role of the CSDn still remains elusive. Earlier studies proposed a polarity in the CSDn's

neurites and suggested that it integrates input from higher protocerebral circuits to modulate the AL in a top-down manner (*Dacks et al., 2009*; *Hill et al., 2002*; *Sun et al., 1993*; *Roy et al., 2007*). Our data is more consistent with a recent suggestion that the CSDn forms local circuits operating at the level of each glomerulus (*Kloppenburg and Mercer, 2008*). We found both pre- and postsynaptic specializations throughout virtually all glomeruli in the AL and confirmed that these connections are functional. Additionally, we show cellular correlates of both synaptic inputs and outputs in CSDn arbors within the LH (*Figure 3*, *Figure 3—figure supplement 3*). Together this anatomical and physiological evidence argues at minimum that top-down modulation is not the exclusive function of the CSDn.

Within the AL, we found that the CSDn makes connections onto GABAergic LNs and most PNs. Both LNs and PNs are inhibited by serotonin from the CSDn and are indirectly excited by Ach during CSDn stimulation. Our immunohistochemistry assays suggest that the CSDn does likely release Ach. We also observedthat blocking 5-HT transmission with methysergide has no effect on the postsynaptic depolarization of the LNs and PNs. Thus 5-HT cannot be the only means by which the CSDn communicates with AL neurons. However, the identity of the neurons that are monosynaptically connected to the CSDn via Ach remains unknown. Combining our TERPS strategy with calcium imaging and pharmacology should provide a powerful means to identify and disambiguate all of the serotonergic and cholinergic postsynaptic partners of the CSDn.

As mentioned above, CSDn stimulation alone was not sufficient to induce strong, lasting modulation in pheromone sensitive glomeruli. So what might be the function of CSDn connectivity within the AL? One possibility is that the CSDn robustly modulates non-pheromone channels. We performed a pilot study of randomly patched PNs from the dorsal medial cluster, but we never observed strong CSDn-mediated modulation (data not shown). An alternative hypothesis is that the CSDn might make odor responses more reliable across those glomeruli that it densely innervates. Functionally, the CSDn hyperpolarizes AL neurons through 5-HT and excites them in an Ach dependent manner. Interestingly, CSDn-evoked activity in PNs and LNs are virtually indistinguishable. At resting membrane potentials these synapses are quite weak as measured at the soma. Importantly though, the effect of CSDn stimulation became pronounced when LNs were depolarized by current injection and firing in LNs was suppressed via strong serotonergic inhibition (*Figure 3*). CSDn stimulation also depolarized LNs more strongly when LNs where hyperpolarized from their resting potentials. This is consistent with the notion that the whichever active current is furthest from its reversal potential will dominate the net effect on the membrane potential. Thus CSDn spiking should serve to narrow the range of LN and PN activity, reduce membrane fluctuations arising from noise, and promote all odor responses to be initiated from a similar baseline. Subsequently, during the odor stimulus, the CSDn is inhibited through reciprocal synapses from LNs, and AL neurons are free to fire in a manner dictated by the odor itself. Such a configuration should reduce the variability in the absolute value of PN and LN responses across different presentations of the same olfactory stimulus resulting in more reliable olfactory coding.

## Serotoninergic function in mammalian and *Drosophila* olfactory circuits

The olfactory systems of flies and mice display remarkable similarity in their organization and function (*Kaupp, 2010*; *Wilson, 2013*), and such similarity extends to their modulation by serotonin. Our results are qualitatively similar to recent studies addressing the effects of endogenous serotonin on olfaction in mice (*Petzold et al., 2009*; *Kapoor et al., 2016*). In both cases, serotonin has a predominantly inhibitory effect on the principal neurons within the first olfactory relay, and broadly blocking serotonin receptors boosts odor responses in these same cells. In both mice and flies, serotonin is also likely released with a fast excitatory neurotransmitter, and the net effect of stimulating serotonergic cells on downstream neurons is brief excitation. In both mice (*Kapoor et al., 2016*) and flies, the effects of brief activation of serotonergic neurons on odor responses appears to be mediated by their fast co-transmitter (*Figure 8*). And finally, only strong sustained activation of the entire serotonergic network leads to a long-lasting suppression of odor responses in mice (*Petzold et al., 2009*) and flies.

In mammals, several important questions remain regarding serotonergic modulation in the OB. One such question is what stimuli evoke activity in raphe neurons that specifically project to the OB. Previous studies have reported dynamic odor responses in raphe neurons during olfactory tasks (*Ranade and Mainen, 2009*; *Cohen et al., 2015*), but the axons of these cells could not be traced

back to the OB. It is therefore always possible that their olfactory receptive fields are being generated by input from higher-cortical areas. In one study, a subset of raphe neurons showed significant inhibition during odor sampling(*Ranade and Mainen, 2009*). These cells could serve the analogous function of the CSDn in mice if they indeed project to the OB. A second question that is common to both mice and flies is why robust serotonergic modulation is only observed with strong stimulation of the entire serotonergic network, and importantly, what might cause this type of activity. Given the similarities of serotonergic modulation in the olfactory systems of mice and flies, studies in the CSDn should shed light into the function and organization of serotonergic modulation in mammalian model systems.

# Materials and methods

## Flies

Flies were reared on Nutri-Fly Bloomington Formulation (Flystuff.com, San Diego, CA) at 25°C. All experiments were performed on flies 1–3 days post-eclosion. *Figure 1E* was performed on male and female flies and all other data are from female flies. All fly stocks containing the Chrimson transgene were raised on rehydrated potato flakes (Caroline Biological, Burlington, NC) mixed with all-*trans*-retinal (see below). A list of all genotypes and their sources for each figure is listed in *Table 1*.

**Table 1.** Odors used in the study.

| Odors | Supplier |
| --- | --- |
| 1-hexanol | Sigma-Aldrich CAS: 111-27-3 |
| 1-octanol | Sigma-Aldrich CAS: 111-87-5 |
| 1-pentanol | Sigma-Aldrich CAS: 71-41-0 |
| Acetic acid | Sigma-Aldrich CAS: 64-19-7 |
| Ammonium hydroxide | Sigma-Aldrich CAS: 1336-21-6 |
| Apple cider vinegar | Spectrum Naturals |
| Benzaldehyde | Sigma-Aldrich CAS: 100-52-7 |
| Beta-citronellol | Sigma-Aldrich CAS: 106-22-9 |
| Butyl acetate | Sigma-Aldrich CAS: 123-86-4 |
| Butyric acid | Sigma-Aldrich CAS: 107-92-6 |
| cVA | Pherobank, Wijk bij Duurstede, Netherlands |
| Ethyl acetate | Sigma-Aldrich CAS: 141-78-6 |
| Ethyl propionate | Sigma-Aldrich CAS: 105-37-3 |
| Geosmin | Sigma-Aldrich CAS: 16423-19-1 |
| Limonene | Sigma-Aldrich CAS: 5989-27-5 |
| Linalool | Sigma-Aldrich CAS: 78-70-6 |
| MCH | Sigma-Aldrich CAS: 589-91-3 |
| Methyl laurate | Sigma-Aldrich CAS: 111-82-0 |
| Methyl salicylate | Sigma-Aldrich CAS: 119-36-8 |
| Paraffin oil | J.T.Baker CAS: 8012-95-1 |
| Pentanoic acid | Sigma-Aldrich CAS: 109-52-4 |
| Pentyl acetate | Sigma-Aldrich CAS: 628-63-7 |
| Phenylacetaldehyde | Sigma-Aldrich CAS: 122-78-1 |
| Propyl acetate | Sigma-Aldrich CAS: 109-60-4 |
| Trans-2-hexen-1-al | Sigma-Aldrich CAS: 6728-26-3 |

## Odors and odor delivery

Odors were presented as previously described (*Bhandawat et al., 2007*), with a few notable exceptions. In brief, a carrier stream of carbon-filtered house air was presented at 2.2L/min to the fly continuously. A solenoid was used to redirect 200 ml/min of this air stream into an odor vial before rejoining the carrier stream, thus diluting the odor a further 10-fold prior to reaching the animal. All odors are reported as v/v dilutions in paraffin oil (J.T. Baker VWR #JTS894), except for acids, which were diluted in distilled water. All odors were obtained from Sigma Aldrich (Saint Louis, MO) except for cVA, which was obtained from Pherobank (Wageningen, Netherlands). A complete list of odors used can be found in *Table 2*. cVA and methyl laurate were delivered as pure odorants. In our olfactometer design, the odor vial path was split to 10 channels each with a different odor or solvent control. Pinch valves (Clark Solutions, Hudson MA part number PS1615W24V) were used to select stimuli between each trial. Thus for *Figure 1D*, each odor was presented sequentially one trial at a time. Each odor was presented 4–6 times within a preparation and the mean of these responses were then averaged across animals. For *Figure 1*, An odor was presented every 30 s, but the same odor was never presented twice within 90 s to prevent depletion of the odor vial's headspace. For figures involving only cVA, the odor was presented every 80 s. For *Figure 6D*, seven different vials

**Table 2.** Drosophila genotypes used in the study.

|  | Genotypes (transgene with Bloomington number) |
|---|---|
| *Figure 1–2* | w⁻;; Gal4-R60F02 (48228), UAS-mCD8-GFP |
| *Figure 2E* | UAS-dSerT-GFP (24463); +; Gal4-R60F02 |
| *Figure 2F* | w⁻; UAS-DenMark (33062); Gal4-R60F02 |
| Supplementary for *Figure 2A–C* | w⁻; UAS-DenMark (33062); Gal4-R60F02 |
| Supplementary for *Figure 2D–F* | UAS-dSerT-GFP; +; Gal4-R60F02 |
| *Figure 3A–G* | CS; UAS-Chrimson/+; Gal4-R60F02, UAS-mCD8-GFP/+ |
| *Figure 3H–J* | w⁻; UAS-Chrimson/UAS-NaChBac (9466); Gal4-R60F02, UAS-mCD8-GFP/+ |
| Supplementary *Figure 1– 2* and for *Figure 3* | w⁻; UAS-Chrimson/+; Gal4-R60F02, UAS-mCD8-GFP/+ |
| Supplementary *Figure 3A–D* for *Figure 3* | w⁻; ChAT-Gal4; UAS-GFP (6793) |
| Supplementary *Figure 3E–L* for *Figure 3* | w⁻;; Gal4-R60F02 (48228), UAS-mCD8-GFP |
| Supplementary *Figure 4A–B* for *Figure 3* | w⁻; UAS-Chrimson/UAS-NaChBac; Gal4-R60F02, UAS-mCD8-GFP/+ |
| Supplementary *Figure 4D* for *Figure 3* | w⁻; UAS-Chrimson/Gal4-Orco (26818); UAS-NaChBac/+ |
| *Figure 4A–D* | CS; UAS-Chrimson/+; Gal4-R60F02, UAS-mCD8-GFP/+ |
| *Figure 4E– G* | w⁻; UAS-Chrimson/UAS-NaChBac; Gal4-R60F02, UAS-mCD8-GFP/+ |
| Supplementary for *Figure 4* | w⁻/CS; UAS-Chrimson/+; Gal4-R60F02, UAS-mCD8-GFP/QF-GH146, QUAS-mCD8-GFP (30038) |
| *Figure 5A– E,H* | CS; QF-Mz19 (41573), QUAS-mCD8-GFP (30002)/Cyo |
| *Figure 5I–J* *Figure 5F–G*, | Gal4-NP3062, UAS-mCD8-GFP |
| Supplementary for *Figure 5* | CS; QF-Mz19, QUAS-mCD8-GFP/Cyo |
| *Figure 6A–C* | w⁻/CS; QF-Mz19, QUAS-mCD8-GFP/UAS-Chrimson; Gal4-R60F02, UAS-mCD8-GFP/+ |
| *Figure 6D–F* | CS; QF-Mz19, QUAS-mCD8-GFP/Cyo |
| *Figure 7A–C* | w⁻/CS; QF-Mz19, QUAS-mCD8-GFP/UAS-Chrimson; Gal4-Trh (38389)/+ |
| *Figure 7D–F* | w⁻/CS; UAS-Diphthᵗˢ (25039)/ QF-Mz19, QUAS-mCD8-GFP; Gal4-R60F02, UAS-mCD8-GFP/+ |
| *Figure 7G–I* | w⁻/CS; UAS-Diphthᵗˢ/ QF-Mz19, QUAS-mCD8-GFP; Gal4-Trh/+ |
| *Figure 8* | w⁻/CS; QF-Mz19, QUAS-mCD8-GFP/UAS-Chrimson; Gal4-R60F02, UAS-mCD8-GFP/+ |
| Supplementary for *Figure 8* | w⁻/CS; QF-Mz19, QUAS-mCD8-GFP/UAS-Chrimson; Gal4-R60F02, UAS-mCD8-GFP/+ |

UAS-Chrimson was from Dr. Vivek Jayaraman, Janelia Farm, Ashburn, VA
Gal4-NP3062 was from Dr. Rachel Wilson, Harvard Medical School, Boston, MA
Gal4-R60F02, Gal4-Trh, UAS-Chrimson, QF-Mz19 were backcrossed with Canton-S flies.

of ammonia were utilized, and we cycled our odor presentation through each vial sequentially, again to prevent headspace depletion. We found it critical to flush the olfactometer lines overnight prior to each experiment to prevent odor contamination defined as CSDn responses to empty odor vials or solvent-only control vials. As the CSDn likely integrates input from all LNs (based on our estimation of connectivity probability), it is not surprising that even small amount of residual odor in the olfactometer can create odor responses to solvent controls.

## Electrophysiology

### Whole-cell recordings

In vivo whole-cell recordings were performed as previously described (*Wilson et al., 2004*; *Gaudry et al., 2013*). Data were low-pass filtered at 5 kHz using an AM Systems model 2400 amplifier (AM Systems, Carlsborg, Washington) and digitized at 10 kHz. Pipettes were pulled from thin-walled borosilicate glass (World Precision Instruments, Sarasota, FL; 1.5 mm outer diameter, 1.12 mm inner diameter) to a resistance of 8–12 MΩ. An exception to our previous methodology is that we visualized neurons using oblique illumination from an infrared LED guide through a fiber optic (Thorlabs, Newton, New Jersey) (*Maimon et al., 2010*). The external recording solution contained in mM: 103 NaCl, 3 KCl, 5 $N$-tris(hydroxymethyl)methyl-2- aminoethane-sulfonic acid, 8 trehalose, 10 glucose, 26 $NaHCO_3$, 1 $NaH_2PO_4$, 1.5 $CaCl_2$, and 4 $MgCl_2$ (adjusted to 270–275 mOsm). The saline was bubbled with 95% $O_2$/5% $CO_2$ and reached a pH of 7.3. Our internal solution contained in mM: 140 potassium aspartate, 10 HEPES, 4 MgATP, 0.5 $Na_3$GTP, 1 EGTA, and 1 KCl. For whole cell recordings, a small hyperpolarizing current was applied to offset the depolarization caused by the pipette seal conductance. Their resting potentials were adjusted slightly to match the firing rate of similar neurons obtained in cell-attached recordings. Neurons which did not fire spontaneously or that had depolarized membrane potentials upon break-in were excluded from the study. Cells were held at -60 mV for voltage clamp recordings. For these experiments, the antenna of the fly was removed to minimize potential polysynaptic contributions of the ORNs during CSDn stimulation.

### Cell identification

To identify the CSDn, we recombined the Gal4 promoter line R60F02 (*Singh et al., 2013*; *Jenett et al., 2012*) with a UAS-mcd8-GFP line. This line is referred to as CSDn-Gal4 throughout the manuscript and labels the CSDn and only a small set of additional neurons (*Singh et al., 2013*). These other neurons were never visible under epifluorescence in our recording set up. The CSDn is unambiguously identifiable as the only visible GFP-positive neuron under our epifluorescence microscopes (Zeiss Axioscop with Thorlabs LED model M470L3 and Dage MTI IR-1000 camera). The cell is located on the posterior lateral edge of the antennal lobe and has a large soma size, large action potentials (>30 mV), and a low spontaneous firing rate of 1–2 Hz. Initially, we used biocytin cell fills to confirm this cell to be the CSDn, but we subsequently relied only on GFP labeling, cell size, position, and physiology to confirm its identity in later recordings. DA1 PNs were labeled using the Q/QUAS system (*Potter et al., 2010*) with the MZ19-QF promoter. These cells were identified based on their soma location in the lateral cluster of the antennal lobe and responsiveness to cVA. This allowed us to specifically target DA1 PNs while manipulating the CSDn using the Gal4/UAS system. VA1d was also identified in Mz19-QF, but was located dorsally and medially. This cell did not respond appreciably to cVA, but did respond to methyl laurate. For pharmacological experiments, DM6 and DL5 were identified in the NP3062 promoter line (under Gal4/UAS expression) by their unique odor responses, positions, and size (*Gaudry et al., 2013*). Randomly selected LNs were identified as LNs based on their large soma size and high amplitude action potentials. They are believed to be GABAergic as they were located in the dorsal lateral cluster and had physiology characteristic of GABAergic LNs. Glutamatergic LNs are located ventrally (*Liu and Wilson, 2013*) and eLNs can be identified by their characteristic IPSPs (*Yaksi and Wilson, 2010*). We are thus confident that the LNs that we sampled are GABAergic. Random PNs were selected from the medial dorsal cluster and generally had small soma sizes and action potentials.

## EAG

EAG recordings were used to estimate the total activity of the ORN population. EAGs were conducted as previously described (*Olsen et al., 2010*). A sharp glass microelectrode filled with external recording solution was inserted midway proximal to distal on the ventral aspect of the antenna. A second micropipette was inserted into the eye of the fly to serve as a reference electrode. Odors were presented in an identical manner as our whole-cell recordings.

## TERPS

We expressed NaChBac and Chrimson in the CSDn with the R60F02-Gal4 line. TTX (1 μm) was used to block all spiking in the AL. We confirmed this by depolarizing each neuron and observing that they could no longer fire action potentials. Generally, evoked responses in saline were much larger compared to responses after the addition of TTX, suggesting that polysynaptic connections do exist. Stimulation of CSDns expressing NaChBac resulted in large plateau potentials consistent with the slow inactivation kinetics of the NaChBac channel (*Ren et al., 2001*). In some preparations spontaneous NaChBac plateau potentials could be seen during CSDn recordings. Similarly, in some LN recordings, we observed spontaneous hyperpolarizations that were indistinguishable from CSDn evoked hyperpolarizations. These spontaneous events were also completely blocked by methysergide. This suggests that they are serotonergic and likely originate from the CSDn. The residual depolarization left after mecamylamine and methysergide application was insensitive to $Cd^{++}$ application. We thus believe it to be mediated by electrical coupling. As gap junctions are generally low pass filters, it is reasonable that the broad NaChBac spike might be transmitted more efficiently from the CSDn to postsynaptic cells compared to a burst of typical fast spikes. We also tested this approach in the cVA circuit of the fly by patching third order cVA sensitive neurons in the lateral horn (*Kohl et al., 2013*). Here we found that chronic expression of NaChBac dramatically altered the synaptic connectivity between DA1 PNs and these neurons in the lateral horn such that responses no longer resembled previously published reports. Therefore caution should be exercised when chronically expressing NaChBac to assess connectivity in any system.

## Pharmacology
### Chemicals

The following chemicals were used in this study at the concentrations indicated: methysergide maleate (50 μm, Tochris, CAS 129-49-7), fluoxetine (10 μM, Tocris/Sigma, CAS 56296-78-7), CGP54626 (50 μM, Tocris, CAS 149184-21-4), mecamylamine (100 μM, Sigma, CAS 826-39-1), tetrodotoxin (1 μM, Tocris, CAS 18660-81-6), picrotoxin (5 μM for GABA blockage and 100 μM for glutamate blockage, Sigma, CAS 124-87-8), glutamate (10 mM, Sigma, CAS 6106-04-3), acetylcholine (50 mM, Sigma, CAS 60-31-1), GABA (250 mM, Sigma, CAS 56-12-2), and serotonin (100 μM, Sigma, CAS 153-98-0). Serotonin solutions were made fresh from powder immediately prior to each experiment and wrapped tightly in aluminum foil to prevent oxidation by light (*Dacks et al., 2009*). We used a peristaltic pump to recirculate the external recording solution in all experiments using pharmaceuticals. Typically, drugs were added sequentially to the same recirculating solution and the effects most drugs used in this study were not able to be washed out by flushing saline through the recording chamber.

### Pressure injection

Glutamate, GABA, and acetylcholine were pressure injected into the antennal lobe with a custom-built pressure injector. A small solenoid valve (The Lee Company, LFAA1201610H) was inserted between a filtered house-air line and the suction port of our pipette holder. A sharp microelectrode was then filled with deionized water containing one of the following transmitters, acetylcholine (50 mM), GABA (250 mM), or glutamate (10 mM). The pressure behind the valve was set to 10 psi, and a brief opening of the valve for 20–100 ms was used to deliver the drug. We repeated these experiments in the presence of antagonists to eliminate mechanical stimulation as a possible explanation for our results. If the injection pipette was removed slightly outside the antennal lobe, we did not see responses in the CSDn. This suggests our results are due to injection of the drugs directly into the antennal lobe, and not diffusion to distal sites on the CSDn arbor.

## Optogenetic stimulation

We used a high-powered red LED (Red XP-E, 620–630 nm wavelength) and Buckpuck driver (RapidLED, Randolph, Vermont) to stimulate Chrimson expressing neurons. The LED was mounted directly underneath the preparation and light was presented at 0.238 mW/mm$^2$ as measured by a Thorlabs light meter PM100A with light sensor S130C. Flies expressing Chrimson were raised on food containing 0.2 mM all-trans-retinal. mM all-*trans*-retinal. All-trans retinal was prepared as a stock solution in ethanol (35 mM), and 28 µl of this stock was mixed into approximately 5 ml of rehydrated potato flakes and added to the top of a vial of conventional food. For *Figures 6A*, *7A* and *8A*, we varied the red light intensity as a 10 Hz sine wave from 0 to 0.238 mW/mm$^2$. This phasic stimulation should prevent adaptation of the optogenetic tool.

## Cell ablation

We used the Gal4 UAS system to express a temperature variant of diphtheria toxin in the CSDn and TRH neurons (*Han et al., 2000*). These flies also possessed the Q-Mz19 and QUAS-mcd8-GFP transgenic elements to label DA1 PNs. Ablation with the Gal4 system did not cause the death of DA1 PNs under Q/QUAS control. Flies were raised at 18°C to prevent premature death of these cells. One day post eclosion, the adult flies were transferred to 30°C for three days. The efficiency of the diphtheria toxin was assessed by immunohistochemistry for the serotonin antibody for each preparation post hoc. We imaged at multiple gain levels to confirm that all target neurons were indeed ablated. Serotonergic cells appear bright throughout their soma whereas background staining appears to only stain the outer membrane of non-5-HT cells. Flies expressing diphtheria toxin with the Trh promoter line displayed a high mortality rate of ~70%. Despite this high rate of death, we were able to obtain some viable animals to use for electrophysiology.

## Immunohistochemistry

We used the following primary antibodies at the indicated dilutions: 1:50 mouse anti-bruchpilot (nc82, Developmental Studies Hybridoma Bank, Iowa City, Iowa), 1:50 rat anti-CD8 Invitrogen (Waltham, MA)(MCD0800), 1:100 rabbit anti-5HT Sigma (S5545), 1:50 chicken anti-GFP Invitrogen (A10262), 1:200 rabbit anti-VAchT acbam (AB68984), and 1:200 Goat anti-ChAT EMD Millipore (AB144P). Secondary antibodies from Invitrogen were used at dilutions of 1:250, which were Alexa Fluor 633 goat anti-mouse lgG (Life Technologies, A21050), Alexa Fluor 568 goat anti-mouse lgG (Life Technologies, A11004), Alexa Fluor 633 goat anti-rabbit (Invitrogen, A21071), Alexa Fluor 488 goat anti-rat lgG (Life Technologies, A11006), Alexa Fluor 488 goat anti-chicken (Life Technologies, A11039), and Alexa Fluor 594 donkey anti-goat abcam (AB150136). Blocking serums were donkey serum from Sigma Aldrich (D9663) and goat serum Vector Labs (S-1000).

## Statistical analysis

Two-tailed paired t-tests were performed for all comparisons between one drug treatment and the control within the same group. Two-tailed one sample t-tests were performed in our voltage-clamp analysis to compare the size of currents with the baseline after baseline subtraction (testing against a mean of zero). A repeated measures ANOVA was performed for multiple comparisons between two or more drug treatments and controls within the same group. A one-way ANOVA was performed for *Figure 5E* because each preparation did not receive every treatment. A Tukey-Kramer post-hoc test was performed after each ANOVA.

## Acknowledgements

We thank David Schoppik, Katherine Nagel, Elizabeth Hong, Daniel Butts, and members of the Gaudry Lab for critical comments on this manuscript. We thank Vivek Jayaraman for UAS-Chrimson flies. All confocal microscopy was obtained at the University of Maryland Imaging Core. We also thank the UMD machine shop for parts fabrication. This work was supported by start up funds provided by the University of Maryland and a Whitehall Fellowship to QG.

## Additional information

### Funding

| Funder | Grant reference number | Author |
| --- | --- | --- |
| University of Maryland | Set up Funds | Quentin Gaudry |
| Whitehall Foundation | 4337750 | Quentin Gaudry |

The funders had no role in study design, data collection and interpretation, or the decision to submit the work for publication.

### Author contributions

XZ, QG, Conception and design, Acquisition of data, Analysis and interpretation of data, Drafting or revising the article

### Author ORCIDs

Quentin Gaudry, http://orcid.org/0000-0002-6869-1253

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
