## [Decision Letter]

Thank you for submitting your article "Functional integration of a serotonergic neuron in the *Drosophila* antennal lobe" for consideration by *eLife*. Your article has been reviewed by three peer reviewers, and the evaluation has been overseen by a Reviewing Editor and K VijayRaghavan as the Senior Editor. The following individual involved in review of your submission has agreed to reveal his identity: Stephen D Shea (Reviewer #3).

The reviewers have discussed the reviews with one another and the Reviewing Editor has drafted this decision to help you prepare a revised submission.

Summary:

The authors performed a highly interesting study that elucidates the role of a single pair of serotonergic interneurons (called CSDns) for odor-evoked responses and olfactory processing in the *Drosophila* antennal lobe. By using different electrophysiological methods, such as whole-cell and voltage clamp recordings, combined with optogenetics manipulations, the authors show that these CSDns are mainly inhibited by odor application through reciprocal synapses deriving within the antennal lobe. Moreover, the authors reveal that CSDns release multiple transmitters, namely in addition to serotonin, the excitatory transmitter acetyl choline. Both transmitters seem to have opposing effects and time courses on their target neurons. Finally by performing a series of experiments, the authors provide evidence that this serotonergic network modulates odor-evoked responses of olfactory projection neurons.

A concern is that the paper is written as a series of partial aspects that do not converge well into a coherent picture. Performing a detailed series of experiments on a neuron that finally is not responsible for the function proposed leaves the reader with lots of information that does not build a clear picture. More so because negative results are hard to prove as certain. in part due to the style of the writing and presentation of the data, and in part due to the confusing and conflicting nature of the results themselves, it is not easy to extract a coherent picture of how this neuron contributes to antennal lobe processing and odor perception. Since the experiments are carefully executed and there are several interesting observations, we are open to seeing the data presented in a more effective manner. As it stands, the manuscript does not always flow well and has several conceptual cul-de-sacs and non-sequiturs.

In sum though, this is a interesting study. We have several specific concerns, in addition to the important general ones above that need the authors' attention. Together, these could substantially improve the manuscript and we hope that this will result in a more compellingly written paper.

Essential revisions:

How specific is the line R60F02-GAL4 that the authors used in their study to label specifically serotonergic neurons in the antennal lobe (This has been addressed other studies and could be explicitly stated)?

It would be preferable to show an immuno staining of the whole brain and not just a cropped image as shown in Figure 1

- What is the n-value for the recordings in Figure 1? Please add that info to the Figure legend.

- For clarity please color the odor names accordingly to the chemical classes in Figure 1.

- the authors state that they included all odor ligands for all known Ors. Why wasn't geosmin included?

- Please add quantification over several animals and statistics to the data in Figure 2. Otherwise it is difficult to judge the reliability and variability of the data shown.

- Please add an image of the whole brain, and not just the antennal lobes, in Figure 2 to show that there is not further labeling in the brain. Since the labeling is rather sparse (even in the antennal lobe) we would suggest showing a z-projection instead of a single focal plane to give the reader an impression of the complete staining. Which glomeruli are innervated and labeled by the CSDns? Do pre- and postsynaptic sites overlap? Do some glomeruli receive only presynaptic input, while others reveal only postsynaptic sites?

- Please add quantification over several animals and statistics to the data of Figure 3.

- How have the LNs in Figure 5 being selected? Did the authors use a LN-specific GAL4 line? Are those GABAergic or glutamatergic?

- The authors should use a repeated measures ANOVA in Figure 6 instead of a one-tailed paired T-Test, since the data are dependent from each other and more than two comparisons are done.

Which statistical test was used in Figure 8? The authors mentioned once a one-tailed t-test. However, a two-tailed t-test has to be used in every case (this is also valid for Figure 9, in case a one-tailed test was used). Please be more precise regarding the statistics that you are applying. This is very difficult to follow throughout the whole MS.

The manuscript needs a thorough proofreading. Readability suffers from a number of typos. E.g. where the authors mean 'principal,' they use 'principle' throughout.

The authors are generally too slavish and literal in comparing their results to the 5HT system of rodents. Some of this is of course appropriate, but they should allow that there may be major differences in the fly and let their data stand more on their own.

Regarding the authors comment on the excitatory input to CSDns that is unmasked by blocking odor-evoked inhibition (Figure 2). It is not blocked by AL applied glutamate antagonists, so do they have any ideas about its source?

Schematics in 5A and 6A depict apparent direct connections from the CSDns to LNs and PNs. Is it known that both direct synapses exist? The evoked currents are pretty small, and is it weird that the traces from both cell types are virtually indistinguishable? Taking these results at face value, we infer that there is a direct connection from CSDNs to PNs and an indirect connection via LNs. Could this circuit complexity explain some of the conflicting results between endo and exo 5HT in addition to the cotransmission?

Related, in some sense the cotransmission of ACh in CSDns is their most interesting result, yet it is not reflected in the title or abstract, and it seems to get dropped before the end of the results. This contributes to narrative confusion.

The most confusing aspect of the study is the apparent difficulty with demonstrating that CSDNs contribute any functional modulation of PN output. The authors spend a lot of time on negative results at this point in the manuscript, leading to an increasingly confused reader. Why did the authors choose to activate CSDns with ammonia before optogenetic activation? In fact, all methods that selectively activate the CSDNs fail to modulate PNs. And yet a 5HT antagonist and a 5HT reuptake inhibitor do modulate them. Also, activating the whole 5HT system modulates PN output. The authors attribute that to some facilitation of CSDn efficacy by other 5HT neurons. While that is possible, they are not able to test whether the CSDns are even necessary for the effects of activating the whole 5HT system. How certain is it that CSDns are the sole source of AL 5HT in flies? While these conflicting results could be simply explained if there were an alternative source of 5HT to the AL the results of Roy et al. and Dacks et al. suggest otherwise. So, these nettles should be grasped in the discussion and addressed.

---

## [Author Response]

*Summary:*

*The authors performed a highly interesting study that elucidates the role of a single pair of serotonergic interneurons (called CSDns) for odor-evoked responses and olfactory processing in the Drosophila antennal lobe. By using different electrophysiological methods, such as whole-cell and voltage clamp recordings, combined with optogenetics manipulations, the authors show that these CSDns are mainly inhibited by odor application through reciprocal synapses deriving within the antennal lobe. Moreover, the authors reveal that CSDns release multiple transmitters, namely in addition to serotonin, the excitatory transmitter acetyl choline. Both transmitters seem to have opposing effects and time courses on their target neurons. Finally by performing a series of experiments, the authors provide evidence that this serotonergic network modulates odor-evoked responses of olfactory projection neurons.*

*A concern is that the paper is written as a series of partial aspects that do not converge well into a coherent picture. Performing a detailed series of experiments on a neuron that finally is not responsible for the function proposed leaves the reader with lots of information that does not build a clear picture. More so because negative results are hard to prove as certain. in part due to the style of the writing and presentation of the data, and in part due to the confusing and conflicting nature of the results themselves, it is not easy to extract a coherent picture of how this neuron contributes to antennal lobe processing and odor perception. Since the experiments are carefully executed and there are several interesting observations, we are open to seeing the data presented in a more effective manner. As it stands, the manuscript does not always flow well and has several conceptual cul-de-sacs and non-sequiturs.*

*In sum though, this is a interesting study. We have several specific concerns, in addition to the important general ones above that need the authors' attention. Together, these could substantially improve the manuscript and we hope that this will result in a more compellingly written paper.*

We thank the reviewers for their comments and critiques of our work. We have rewritten large portions of the manuscript for better flow and to more clearly communicate our results. We focused on the transitions between sections to better tie them together and to try to prevent the feeling of "cul-de-sacs". Specifically, we refocused much of the manuscript around pheromone processing and the DA1 glomerulus and we leave open the possibility that other olfactory channels might be modulated more strongly by the CSDn. We still believe the interaction between the CSDn and this glomerulus is of critical importance as cVA processing and the DA1 glomerulus are critical for proper courtship behavior (a widely studied model system). Unfortunately, we tried to develop a LexA-CSDn line to stimulate the CSDn and label other non-pheromone PNs with the Gal4/UAS system, but this new line did not label the CSDn.

We also received feedback that the voltage-clamp analysis bogged down several readers and hindered the pace of the manuscript. We've replaced this entirely with a current clamp analysis that is far simpler to understand. We've kept the voltage clamp data in supplementary figures for two important reasons 1) it doubles the size of our data set, which is important for randomly sampled cells, and 2) we used more flies in the voltage clamp studies to alternate which antagonist was administered first. Adding drugs sequentially to a recording solution is common approach in physiology and many drugs cannot be easily washed out (like methysergide).

We have also drastically revamped the Discussion section to reduce speculation and highlight many of the concerns that were raised by the reviewers. We hope these changes will make the manuscript easier to read and comprehend.

*Essential revisions:*

*Important specific concerns*

*How specific is the line R60F02-GAL4 that the authors used in their study to label specifically serotonergic neurons in the antennal lobe (This has been addressed other studies and could be explicitly stated)?*

The R60F02- Gal4 line labels the CSDn and only a few other neurons. These other neurons appear to project to the subesophageal ganglion and not to the antennal lobe. They are labeled much more weakly than the CSDn and are never visible at our patch-clamping rigs. We only observe these cells on occasion during confocal imaging and none of these other neurons are labeled with the 5-HT antibody. Gal4 expression is often graded and we find the CSDn to be by far the most strongly labeled cell in this line. We thus attribute our findings to the CSDn. We reference Singh et al. 2013 regarding the specificity of the R60F02-Gal4 line. Our observations regarding the expression patterns of R60F02-Gal4 are largely in agreement with their publication with the exception that we saw less labeling in VA1d. In our hands, this was true for labeling with mcd-GFP, dSERT, DenMark, and n-syb. This is the only minor discrepancy that we observed.

It would be preferable to show an immuno staining of the whole brain and not just a cropped image as shown in Figure 1

We have added an image of the whole brain.

- What is the n-value for the recordings in Figure 1? Please add that info to the Figure legend.

This is a raster plot from a single CSDn. The x-axis represents time and the different rows represent subsequent presentations of the same stimulus. We have described the nature of the raster more in the figure legend. This type of data was compiled from 10 CSDns to generate panel E. A raster plot is one method to show "raw" data across many presentations of different stimuli.

- For clarity please color the odor names accordingly to the chemical classes in Figure 1.

We have implemented the suggestion.

*- the authors state that they included all odor ligands for all known Ors. Why wasn't geosmin included?*

We have added geosmin and a number of other ethologically relevant odorants that the reviewer might find interesting. These include methyl laurate and limonene. We also tested geranyl acetate (only one sample at this time and thus not in our figure). This is of interest because it was recently shown using EM that the ORNs for this glomerulus directly synapse onto the CSDn in *Drosophila* larvae (Berck et al. 2016 *eLife*). All of these odors including geranyl acetate inhibit the CSDn. Thus even if the CSDn receives monosynaptic input from a few ORNs, it is still likely that activation of those ORNs will inhibit the CSDn, as long as the same odor recruits enough LN activity. The EAG response of an odor appears to be the strongest predictor of the CSDn response.

- Please add quantification over several animals and statistics to the data in Figure 2. Otherwise it is difficult to judge the reliability and variability of the data shown.

We have done this.

- Please add an image of the whole brain, and not just the antennal lobes, in Figure 2 to show that there is not further labeling in the brain. Since the labeling is rather sparse (even in the antennal lobe) we would suggest showing a z-projection instead of a single focal plane to give the reader an impression of the complete staining. Which glomeruli are innervated and labeled by the CSDns? Do pre- and postsynaptic sites overlap? Do some glomeruli receive only presynaptic input, while others reveal only postsynaptic sites?

We have added images of the whole brain and of the lateral horn in a supplementary figure. We have replaced the original DenMark image with a new z-stack showing stronger labeling. We see that all glomeruli are clearly labeled with both dSerT and DenMark, though some glomeruli do receive more innervation than others. There is a large amount of overlap between staining for these two markers. In short, anywhere that we see mcd8-GFP staining, we also see both DenMark and dSerT staining. Quantitating the overlap between dSerT and DenMark is challenging because DenMark staining is known to result in extremely high levels of background staining (Kohl et al. 2014).

We obtained several large data sets to try to optimize our immunohistochemistry for further quantification. This proved difficult as using antibodies often lead to high levels of non-specific binding (especially for DenMark). Endogenous fluorescence (again, especially for DenMark) was often too dim to see after processing the nc82-channel. In the end we acquired a suitable data set but struggled immensely to identify the 50 glomeruli without specific markers. This is not something anyone in my laboratory has prior experience with. A recent phone conversation with Dr. Andrew Dacks at the University of West Virginia revealed that he has performed the exact analysis that the reviewer is requesting and with better genetic tools to replace DenMark. He is currently writing up this work for publication. As Dr. Dacks is a career insect neuroanatomist, we feel it makes more sense for him to perform this analysis.

Previous models of the CSDn have virtually all claimed it functions in top-down modulation from the protocerebrum back to the AL. We simply intended to use this anatomical data as evidence that both the AL and the lateral horn (LH) are input and output sites for the CSDn.

- Please add quantification over several animals and statistics to the data of Figure 3.

We have replaced this figure with a new current clamp analysis that we believe will be easier for the audience to digest. This new figure replaces Figure 3 and moves the more complicated voltage clamp experiments to a supplementary figure. All new analyses clearly state the n (animal) values and statistics used.

- How have the LNs in Figure 5 being selected? Did the authors use a LN-specific GAL4 line? Are those GABAergic or glutamatergic?

These LNs were targeted without the use of a promoter line. This is due to the lack of a suitable CSDn-LexA line (which we tried to generate, but it was too weak to express GFP or Chrimson). However, the LNs we sampled are almost certainly GABAergic. Glutamatergic LNs are located in the most ventral region of the antennal lobe and are inaccessible in our preparation. To access Glu LNs, we must rotate the head of the fly 180° and patch from the ventral side (Liu and Wilson, 2013). Additionally, GABA and Glu LNs can be distinguished physiologically as Glu LNs have much smaller action potential amplitudes. Acetylcholinergic eLNs are also located more ventrally and have a very characteristic physiological hallmark of continuous barrages of IPSPs (Yaksi and Wilson, 2010). Thus we are confident that the soma location, spike amplitude, and subthreshold activity are sufficient to classify these neurons as GABAergic LNs. However, we certainly do acknowledge that even these GABAergic LNs likely represent a very diverse population, and that our approach must result in a sampling bias towards only dorsal lateral GABA LNs.

- The authors should use a repeated measures ANOVA in Figure 6 instead of a one-tailed paired T-Test, since the data are dependent from each other and more than two comparisons are done.

Which statistical test was used in Figure 8? The authors mentioned once a one-tailed t-test. However, a two-tailed t-test has to be used in every case (this is also valid for Figure 9, in case a one-tailed test was used). Please be more precise regarding the statistics that you are applying. This is very difficult to follow throughout the whole MS.

We have tried to clarify the statistics through the figures to be more consistent and we now include a statistics section in our methods section.

*The manuscript needs a thorough proofreading. Readability suffers from a number of typos. E.g. where the authors mean 'principal,' they use 'principle' throughout.*

We apologize for these errors and have tried to eliminate them from the resubmitted manuscript.

*The authors are generally too slavish and literal in comparing their results to the 5HT system of rodents. Some of this is of course appropriate, but they should allow that there may be major differences in the fly and let their data stand more on their own.*

We have eliminated nearly all reference to mammalian literature in the Results section of the manuscript. We include a single concise section in the Discussion comparing our findings to the results seen in rodents. This should also help improve readability of the manuscript. As no studies prior to ours have ever performed physiology on these cells in flies, we drew much or our experimental inspiration from recently published work in mice.

Regarding the authors comment on the excitatory input to CSDns that is unmasked by blocking odor-evoked inhibition (Figure 2). It is not blocked by AL applied glutamate antagonists, so do they have any ideas about its source?

Because acetylcholine is the primary excitatory neurotransmitter used by the fly, we would presume it is coming from an unidentified acetylcholinergic neuron. Glutamate in the AL tends to be inhibitory (Liu and Wilson, 2013). The two best candidates are the ORNs themselves or the PNs. Recent EM work in the larvae has shown the CSDn to receive monosynaptic input from only two ORN classes. It is possible that once inhibition is blocked in the AL, the excitation from these ORN classes is revealed. Additionally, we recorded from the CSDn while stimulating most PNs with channelrhodopsin (Q-GH146 > QUAS-ChR2) and found that activation of the PNs can cause a small depolarization in the CSDn. We believe any source of excitation is simply masked when inhibition is intact.

*Schematics in 5A and 6A depict apparent direct connections from the CSDns to LNs and PNs. Is it known that both direct synapses exist? The evoked currents are pretty small, and is it weird that the traces from both cell types are virtually indistinguishable? Taking these results at face value, we infer that there is a direct connection from CSDNs to PNs and an indirect connection via LNs. Could this circuit complexity explain some of the conflicting results between endo and exo 5HT in addition to the cotransmission?*

Testing monosynapticity in this circuit has proven exceptionally challenging precisely because the synapses are so small and likely located distally from our somatic recording site. Spike-triggered averages from dual recordings failed to reveal monosynaptic connections between these cells presumably because single-spike-evoked potentials are small. To see any effect postsynaptically, we have to drive a lot of activity in the CSDn, which inherently recruits polysynaptic circuits. These connections were actually presumed from earlier work in flies and moths that suggested that the CSDn synapses onto LNs and PNs. To address the reviewer and to test these connections directly, we developed a new strategy for assessing monosynaptic connections. This approach should work in any model system. In short we block all activity in the brain with tetrodotoxin and rescue spiking selectively in the CSDn through the expression of the TTX-insensitive sodium channel, NaChBac. This approach worked successfully at the cholinergic ORN to LN synapse and for all 5-HT synapses from the CSDn. It did however reveal that all of the acetylcholine effects from the CSDn onto the LNs and PNs are likely polysynaptic. However, it remains certain that the CSDn must still communicate within the AL through a means other than 5-HT alone. This is because methysergide does not block the excitation of LNs and PNs during CSDn stimulation in saline. Thus the CSDn must still either use Ach in the AL or communicate with cholinergic neurons via gap junction coupling. The identity of the monosynaptic cholinergic partners of the CSDn remain unknown.

We are not clear on how the circuit wiring itself could result in the differences that we observe between exogenous applications of serotonin versus endogenous release. We reason that the hard wiring of the circuit would remain the same in both instances and thus both methods of application should have the same effect. We believe that the differences are likely seen do to either the overall concentration of 5-HT or the accessibility of 5-HT to extrasynaptic receptors. Gasque et al. 2013 showed that the fly 5-HT7 receptor is fully saturated prior to concentrations where 5-HT1B receptors even begin to activate. These two receptors have opposite effects on cAMP and have both been implicated in the AL. The exogenous concentrations of 5-HT used in these studies are generally quite high and may simply saturate one or both receptor types in a manner that is never achieved but exogenous release.

*Related, in some sense the cotransmission of ACh in CSDns is their most interesting result, yet it is not reflected in the title or abstract, and it seems to get dropped before the end of the results. This contributes to narrative confusion.*

We tried previously to address the potential roles for Ach and 5-HT in the Discussion. We emphasize this more in the new manuscript. Our new results using NaChBac, however, do call into question whether the CSDn is truly cholinergic. The evidence that the CSDn is cholinergic now relies more heavily on immunohistochemistry and labeling by the ChAT-Gal4 promoter line, but now lacks physiological evidence. We thus now refer to the CSDn as putatively cholinergic.

*The most confusing aspect of the study is the apparent difficulty with demonstrating that CSDNs contribute any functional modulation of PN output. The authors spend a lot of time on negative results at this point in the manuscript, leading to an increasingly confused reader. Why did the authors choose to activate CSDns with ammonia before optogenetic activation? In fact, all methods that selectively activate the CSDNs fail to modulate PNs. And yet a 5HT antagonist and a 5HT reuptake inhibitor do modulate them. Also, activating the whole 5HT system modulates PN output. The authors attribute that to some facilitation of CSDn efficacy by other 5HT neurons. While that is possible, they are not able to test whether the CSDns are even necessary for the effects of activating the whole 5HT system. How certain is it that CSDns are the sole source of AL 5HT in flies? While these conflicting results could be simply explained if there were an alternative source of 5HT to the AL the results of Roy et al. and Dacks et al. suggest otherwise. So, these nettles should be grasped in the discussion and addressed.*

We understand the reviewers' confusion and frustration. We believe that the CSDn is the sole serotonergic neuron to physically innervate the AL because when we kill this neuron with diphtheria toxin, we see no 5-HT processes within in the AL. Our anatomical results in this regard are in direct agreement with Roy et al. 2007 and Dacks et al. 2006. Dacks et al. 2009 used only exogenous serotonin for their physiology and our results with exogenous serotonin are in agreement with that study as well.

There are two ways we can explain how the DA1 glomerulus might be sensitive to serotonergic pharmacology and Trh stimulation, but not to CSDn activation. The first explanation is that serotonin from non-CSDn neurons may modulate downstream neurons that feedback into the AL and into the DA1 glomerulus. There are known projections for other studies that show higher order feedback into the AL. The second possible mechanism is that the DA1 glomerulus may be sensitive to 5-HT in the haemolymph or extracellular space and that serotonin may reach the antennal lobe from outside via paracrine signaling. Our data is consistent with either of these models. Importantly though, our data refute a simple model that the CSDn is responsible for all olfactory modulation. We propose that this may hold true for all olfactory channels (including VA1d), and this should be of general interest in fly olfaction.